EMBO
Molecular Medicine

# Disease modeling of a mutation in α-actinin 2 guides clinical therapy in hypertrophic cardiomyopathy

Maksymilian Prondzynski[1,2,†,‡], Marc D Lemoine[1,2,3,‡], Antonia TL Zech[1,2], András Horváth[1,2,¶], Vittoria Di Mauro[4,5], Jussi T Koivumäki[6], Nico Kresin[1,2], Josefine Busch[1,2], Tobias Krause[1,2], Elisabeth Krämer[1,2], Saskia Schlossarek[1,2], Michael Spohn[7], Felix W Friedrich[1,2], Julia Münch[3,8], Sandra D Laufer[1,2], Charles Redwood[9], Alexander E Volk[10], Arne Hansen[1,2], Giulia Mearini[1,2], Daniele Catalucci[4,5], Christian Meyer[2,3], Torsten Christ[1,2], Monica Patten[2,8], Thomas Eschenhagen[1,2] & Lucie Carrier[1,2,*] [iD]

## Abstract

**Hypertrophic cardiomyopathy (HCM) is a cardiac genetic disease accompanied by structural and contractile alterations. We identified a rare c.740C>T (p.T247M) mutation in *ACTN2*, encoding α-actinin 2 in a HCM patient, who presented with left ventricular hypertrophy, outflow tract obstruction, and atrial fibrillation. We generated patient-derived human-induced pluripotent stem cells (hiPSCs) and show that hiPSC-derived cardiomyocytes and engineered heart tissues recapitulated several hallmarks of HCM, such as hypertrophy, myofibrillar disarray, hypercontractility, impaired relaxation, and higher myofilament Ca$^{2+}$ sensitivity, and also prolonged action potential duration and enhanced L-type Ca$^{2+}$ current. The L-type Ca$^{2+}$ channel blocker diltiazem reduced force amplitude, relaxation, and action potential duration to a greater extent in HCM than in isogenic control. We translated our findings to patient care and showed that diltiazem application ameliorated the prolonged QTc interval in HCM-affected son and sister of the index patient. These data provide evidence for this *ACTN2* mutation to be disease-causing in cardiomyocytes, guiding clinical therapy in this HCM family. This study may serve as a proof-of-principle for the use of hiPSC for personalized treatment of cardiomyopathies.**

**Keywords** disease modeling; human-induced pluripotent stem cells; hypertrophic cardiomyopathy; long QT syndrome; precision medicine

**Subject Categories** Cardiovascular System; Genetics, Gene Therapy & Genetic Disease; Stem Cells & Regenerative Medicine
See also: **A Goedel et al** (December 2019)

## Introduction

Hypertrophic cardiomyopathy (HCM) is an inherited myocardial disease with a prevalence of 1:500 in the general population (Maron *et al*, 1995). It is defined by hypertrophy, mainly seen in the left ventricle (LV) and associated with myocardial disarray, fibrosis, and diastolic dysfunction (Ho, 2010). Patients with HCM exhibit frequent episodes of atrial fibrillation (22%), non-sustained ventricular tachycardia (31%), and annual incidence of sudden cardiac death (1%) (Adabag *et al*, 2005).

There is no curative treatment for HCM, which reverses or prevents hypertrophy and dysfunction of the heart (Tardiff *et al*, 2015). Available drug-based therapies aim at relieving HCM-associated symptoms and decelerating disease progression. β-adrenoceptor antagonists (β-blockers) and Ca$^{2+}$-channel blockers are used to reduce heart rate and therefore lengthen LV filling time (Marian, 2009). While these therapeutic approaches address general disease mechanisms independent of the underlying pathophysiology,

1   Institute of Experimental Pharmacology and Toxicology, University Medical Center Hamburg-Eppendorf, Hamburg, Germany
2   DZHK (German Centre for Cardiovascular Research), partner site Hamburg/Kiel/Lübeck, Hamburg, Germany
3   Department of Cardiology-Electrophysiology, University Heart and Vascular Center, Hamburg, Germany
4   Institute of Genetics and Biomedical Research, Milan Unit, National Research Council, Milan, Italy
5   Humanitas Clinical and Research Center, Rozzano, Milan, Italy
6   Faculty of Medicine and Health Technology, Tampere University, Tampere, Finland
7   Bioinformatics Core, University Medical Center Hamburg-Eppendorf, Hamburg, Germany
8   Department of General and Interventional Cardiology, University Heart and Vascular Center, Hamburg, Germany
9   Radcliffe Department of Medicine, University of Oxford, Oxford, UK
10  Institute of Human Genetics, University Medical Center Hamburg-Eppendorf, Hamburg, Germany
    *Corresponding author. Tel: +49 40 7410 57208; E-mail: l.carrier@uke.de
    †Present address: Department of Cardiology, Boston Children's Hospital, Harvard Medical School, Boston, MA, USA
    ‡These authors contributed equally to this work
    ¶Correction added on Dec 6 2019 after first online publication: The affiliation footnotes from András Horváth were corrected from "2,3" to "1,2".

knowledge of the disease-causing mutation(s) and related cardiomyocyte (CM)/heart function could theoretically inform about specific individual treatment options, the concept of personalized medicine.

Over 1,500 mutations in at least 11 different genes, encoding components of the cardiac sarcomere, have been identified as a potential cause for HCM (Marian & Braunwald, 2017), highlighting the need for precision medicine as genetic diversity of HCM-affected individuals is paralleled by clinical heterogeneity. Furthermore, the causative role of many gene mutations remains controversial in HCM, one example being *ACTN2*, which encodes α-actinin 2. The α-actinin 2 protein is located at the Z-disk of the sarcomere in homodimers fulfilling three main functions: sarcomere formation, anchoring/crosslinking of actin thin filaments, and interaction with titin (Gautel & Djinovic-Carugo, 2016; Chopra *et al*, 2018). Several *ACTN2* variants have been associated with HCM, but only a few could be validated as disease-causing with altered function (Theis *et al*, 2006; Chiu *et al*, 2010; Girolami *et al*, 2014; Haywood *et al*, 2016). One of these studies revealed reduced F-actin-binding affinity, altered Z-disk localization and dynamic behavior of two α-actinin 2 mutants after gene transfer in CMs (Haywood *et al*, 2016).

Here, we identified a novel *ACTN2* mutation (c.740C>T; p.T247M) in a patient with HCM and set out to study its consequences on cellular function, using human-induced pluripotent stem cell (hiPSC)-derived CMs and CRISPR/Cas9 gene editing, an approach increasingly employed to study disease mechanisms in cardiomyopathies (Prondzynski *et al*, 2017; Mosqueira *et al*, 2018; Eschenhagen & Carrier, 2019). We also used engineered heart tissues (EHTs) to study contractile and electrophysiological function in a more physiological and mature 3D format (Uzun *et al*, 2016; Lemoine *et al*, 2017, 2018).

## Results

### Identification of a novel *ACTN2* mutation in an HCM proband and family

DNA sequencing of 16 HCM-associated genes (Appendix Table S1) revealed an *ACTN2* c.740C>T (p.T247M) variant in a HCM patient (II.4; Fig 1A), who presented with LV hypertrophy, LV outflow tract (LVOT) obstruction, atypical atrial flutter, and paroxysmal atrial fibrillation. This variant is not listed in the genome aggregation database (gnomAD), and different *in silico* prediction tools (CADD, REVEL, M-CAP) revealed high pathogenicity scores. The HCM-affected son (III.4) and sister (II.5) of the index patient carried this *ACTN2* mutation, but not the nieces (III.1, III.2, III.3; Fig 1A).

The index patient II.4 underwent myectomy at the age of 58, a surgical ablation and mitral valve repair at 59 years, and a mitral valve replacement at age 64. After myectomy, she developed a left bundle branch block. Atypical flutter was ablated when she was 65 years old. Patient II.5 suffered from paroxysmal supraventricular tachycardia since adolescence. At the age of 56 years, she was diagnosed with HCM, and an electrophysiological examination revealed an atrioventricular nodal reentrant tachycardia, which was treated by slow pathway modulation. She underwent pulmonary vein isolation for paroxysmal atrial fibrillation at the age of 59 and 61 years. Patient III.4 was diagnosed with HCM at the age of 43 in the course of this study. Long-term ECG analysis revealed a low number of premature atrial contractions and polymorphic premature ventricular contractions as single events and as couplets. Cardiac magnetic resonance imaging showed a normal left ventricular function, a moderate septal hypertrophy of 14 mm with an apical pronunciation and without LVOT obstruction. Late gadolinium enhancement appeared in the anterior septum at the insertion to the right ventricle. The half-sister (II.2) of the index patient was diagnosed with HCM at the age of ~55 years with LVOT obstruction. She underwent transcoronary ablation of septum hypertrophy at the age of 59 years, developed supraventricular premature beats, paroxysmal atrial fibrillation, apoplexy, and died at the age of 70 years after post-operative sepsis more than 10 years ago in an external clinic before the investigation of this study. Interestingly, supraventricular premature beats were reported to be reduced by verapamil, but not by bisoprolol. Clinical characterization of affected versus non-affected family members is shown in Table 1. In this HCM family, there is no hint for an early manifestation of HCM and none of the family members reported an early unexpected sudden death in the family.

### Patient-derived hiPSC-CMs exhibit the same level of α-actinin 2 than CRISPR/Cas9-generated control

In order to characterize this rare *ACTN2* variant *in vitro*, dermal fibroblasts from the index patient (II.4) were reprogrammed to hiPSCs (HCM) and used to generate the isogenic control cell line (HCMrep) with the CRISPR/Cas9 nickase approach and homology-directed repair with an exogenous template containing the WT *ACTN2* sequence (Fig 1B, Appendix Fig S1A). Both cell lines were compared to hiPSC derived from a donor individual (Ctrl). As none

---

**Figure 1. HCM-affected family and molecular characterization of human tissue and hiPSC-derived CMs from 2D and 3D models.**

A   Pedigree of HCM-affected family carrying a novel *ACTN2* mutation (c.740C>T; T247M). Individuals who were genotyped are marked with an apostrophe, and HCM patients are indicated by filled symbols and carriers of this *ACTN2* mutation with a (+) sign. Encircled in red with the arrow is the index patient (II.4) from whom the HCM hiPSC line was created.

B   Representative Sanger sequencing results of the *ACTN2* locus in the Ctrl, HCM, and HCMrep hiPSCs (higher magnification of the locus shown on the right, red rectangle). Depicted is the mutation on position g.54,208 (c.740C>T), the repair template from g.54,148 to 54,270 (-strand), the according silent mutations encoded on the repair template on position g.54,200 (G>C; -strand) and g.54,245 (C>G; -strand), and sgRNAs A and B.

C, D   Western blot analysis of hiPSC-CMs (C) and -EHTs (D) stained with antibodies directed against α-actinin 2 and cTnT. Quantification of α-actinin 2 protein levels in single samples, normalized to cTnT and related to Ctrl, is depicted [n = number of samples/differentiations; 2D: Ctrl (n = 4/3); HCM (n = 4/3); HCMrep (n = 4/2); EHT: Ctrl (n = 4/2); HCM (n = 4/3); HCMrep (n = 3/3)]. Data are expressed as mean ± SEM, one-way ANOVA with Bonferroni's post-test. Abbreviations used are as follows: CMs, cardiomyocytes; cTnT, cardiac troponin T; Ctrl, control; EHTs, engineered heart tissues; MW, molecular weight marker.

Source data are available online for this figure.

                                                        

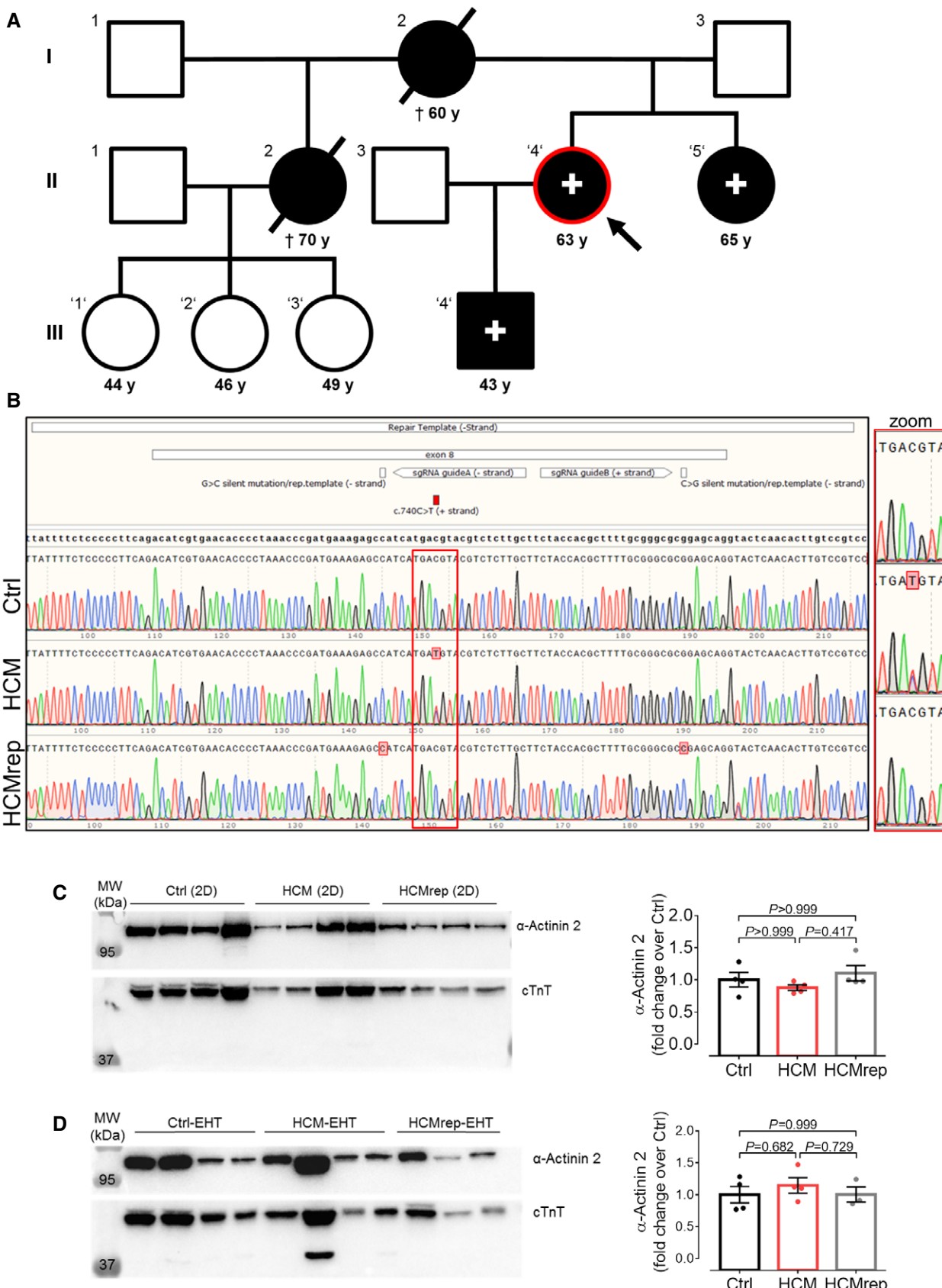

Figure 1.

**Table 1. Clinical characteristics of members of the HCM-affected family.**

| | Ctrl (n = 3) Mean ± SEM | HCM (n = 3) Mean ± SEM | P value |
|---|---|---|---|
| Age (years) | 46.3 ± 1.8 | 57.0 ± 8.6 | 0.211 |
| Body surface (m²) | 2.0 ± 0.1 | 2.1 ± 0.3 | 0.769 |
| Heart rate (bpm) | 84 ± 15 | 76 ± 5 | 0.577 |
| QT (ms) | 359 ± 19 | 406 ± 17 | 0.089 |
| QTcB (ms) | 420 ± 13 | 458 ± 8 | 0.023 |
| QTcF (ms) | 398 ± 1 | 440 ± 8 | 0.003 |
| SBP (ms) | 139 ± 4 | 151 ± 7 | 0.142 |
| DBP (ms) | 85 ± 5 | 82 ± 9 | 0.709 |
| LVEDD (mm) | 39 ± 2 | 45 ± 4 | 0.125 |
| LVESD (mm) | 27 ± 3 | 36 ± 3 | 0.056 |
| LVEDV (ml) | 63 ± 6 | 94 ± 26 | 0.236 |
| LVESV (ml) | 20 ± 1 | 38 ± 13 | 0.180 |
| LVWTmax (mm) | 13.1 ± 0.6 | 17.6 ± 0.1 | 0.001 |
| LVEF (%) | 63 ± 1 | 61 ± 4 | 0.535 |
| Global E′ (cm/s) | 12.9 ± 0.9 | 8.1 ± 1.4 | 0.024 |
| E (cm/s) | 83 ± 8 | 105 ± 27 | 0.384 |
| A (cm/s) | 75 ± 15 | 42 ± 4 | 0.067 |
| E/A | 1.1 ± 0.1 | 2.5 ± 0.5 | 0.025 |
| E/E′ | 6.5 ± 0.2 | 14.3 ± 6.1 | 0.192 |
| Deceleration time (ms) | 116 ± 6 | 206 ± 22 | 0.007 |
| LA vol (ml) | 42 ± 3 | 71 ± 5 | 0.006 |

Three family members as controls (Ctrl), negative (III.1/2/3) for *ACTN2* mutation (c.740C>T; T247M), compared to three family members positive for *ACTN2* mutation (HCM; II-4, II-5, III-4). Data are expressed as mean ± SEM. *P* values derived by unpaired Student's *t*-test; abbreviations used are as follows: QTcB, Bazett-corrected QT; QTcF, Fredericia-corrected QT; SBP, systolic blood pressure; DBP, diastolic blood pressure; LVEDD, left ventricular end-diastolic diameter; LVESD, LV end-systolic diameter; LVEDV, LVED volume; LVESV, LVES volume; LVWTmax, maximum LV wall thickness; LVEF, LV ejection fraction; LA, left atrium; Global E′, early myocardial tissue Doppler relaxation velocity.

of the hiPSC lines presented karyotype abnormalities (Appendix Fig S1B), they were differentiated to CMs (Breckwoldt *et al*, 2017) and subjected to our disease modeling protocol (Appendix Fig S1C). Genotypes were confirmed in genomic DNA from hiPSCs (Fig 1B) and hiPSC-CMs (data not shown).

We then analyzed the α-actinin 2 amount in hiPSC-CMs cultured in 2D and EHT format as well as in LV septum from the index patient II.4. α-Actinin 2 protein levels in hiPSC-CMs did not differ either between lines and culture formats (Fig 1C and D) or between patient II.4 septal myectomy and non-failing heart tissue (Appendix Fig S2A).

To evaluate allelic expression of wild-type (WT) and mutant alleles, we subcloned and sequenced RT–PCR fragments from HCM-EHTs and from the myectomy tissue sample of patient II.4. Analysis of ≥ 13 clones each revealed 48 and 60% mutant clones in EHT and LV septum of the patient, respectively (Appendix Fig S2B), suggesting allelic balance and stable missense transcripts. Allelic balance was confirmed by RNA sequencing (Appendix Fig S2C). Total

transcript levels of *ACTN2* were ~two-fold higher in all HCM samples (hiPSC in 2D, 3D, and human myectomy) than in their corresponding controls (Appendix Fig S2D). Genes related to hypertrophic signaling (*MYH7*, *NPPB*, *SRF*, *FHL1*), Ca²⁺ handling (*CACNA1C*, *PLN*, *RYR2*), fibrosis (*COL1A1*, *CTGF*), and apoptosis (*BCL2*) were dysregulated in HCM myectomy and in HCM hiPSC-CMs or -EHTs (Appendix Fig S2D).

**Patient-derived hiPSC-CMs exhibit HCM phenotypes**

To further evaluate morphological impacts of this novel *ACTN2* mutation, hiPSC-CM lines were seeded at low-density in 96-well plates (Appendix Fig S1C). Immunofluorescence analysis of hiPSC-CMs with an α-actinin 2 antibody showed a cross-striated pattern at 30 days *in vitro* (Fig 2A), indicating proper formation of sarcomeres. Cell area was markedly higher in hiPSC-CMs from HCM than from Ctrl and HCMrep (Fig 2B). Analysis of Z-disk structure showed a higher index of myofibrillar disarray in HCM than in Ctrl and HCMrep (Fig 2C).

To explore functional consequences of the mutation, we casted hiPSC-CM in 3D format as EHTs and analyzed their contractile behavior after 26 days *in vitro* (Fig 2D). Contraction traces of EHTs paced at 1 Hz showed 58 and 19% higher peak force in HCM- than in Ctrl- and HCMrep-EHTs, respectively (Fig 2E and F). Time to peak contraction (T1$_{80\%}$) did not differ between experimental groups (Fig 2G). Relaxation time (T2$_{80\%}$) was 54 and 17% longer in HCM- than in Ctrl- and HCMrep-EHTs, respectively (Fig 2E and H). These data suggest increased contractility and relaxation deficit as an underlying disease phenotype in HCM-EHTs.

To evaluate whether the prolonged relaxation in HCM-EHTs is due to higher myofilament Ca²⁺ sensitivity, a common finding in HCM, we chemically skinned EHTs and human tissue samples and evaluated the force–pCa relationships. In line with the higher peak force in intact HCM-EHTs (Fig 2F), force normalized to cross-sectional area was higher in skinned HCM-EHTs than in Ctrl- or HCMrep-EHTs (Appendix Fig S3A), whereas it did not differ between myectomy of patient II.4 and non-failing heart sample (Appendix Fig S3B). The force–Ca²⁺ relationship was shifted to the left in skinned HCM-EHTs compared to HCMrep and Ctrl-EHTs (Fig 2I; Appendix Fig S3C). Similarly, pCa$_{50}$ was higher in skinned HCM than in non-failing heart muscle strips (Appendix Fig S3D and E). These data suggest that increased myofilament Ca²⁺ sensitivity could contribute to the prolonged relaxation in HCM-EHTs.

**HCM hiPSC-CMs exhibit longer action potential, higher L-type Ca²⁺ channel current density, and reduced binding of α-actinin 2 to L-type Ca²⁺ channels**

Given that the difference in myofilament Ca²⁺ sensitivity was much smaller than the difference in relaxation time between HCM and control cell lines, we further explored electrophysiological consequences of the *ACTN2* mutation. Action potentials (APs) were recorded with sharp microelectrodes in intact EHTs. AP parameters were measured at 1-Hz pacing. There was no difference in take-off potential and upstroke velocity between EHTs from Ctrl, HCM, and HCMrep (Table 2), suggesting similar sodium current function. On the other hand, AP amplitude was higher, and AP duration at 50% (APD$_{50}$) and 90% (APD$_{90}$) of repolarization was ~50% longer in

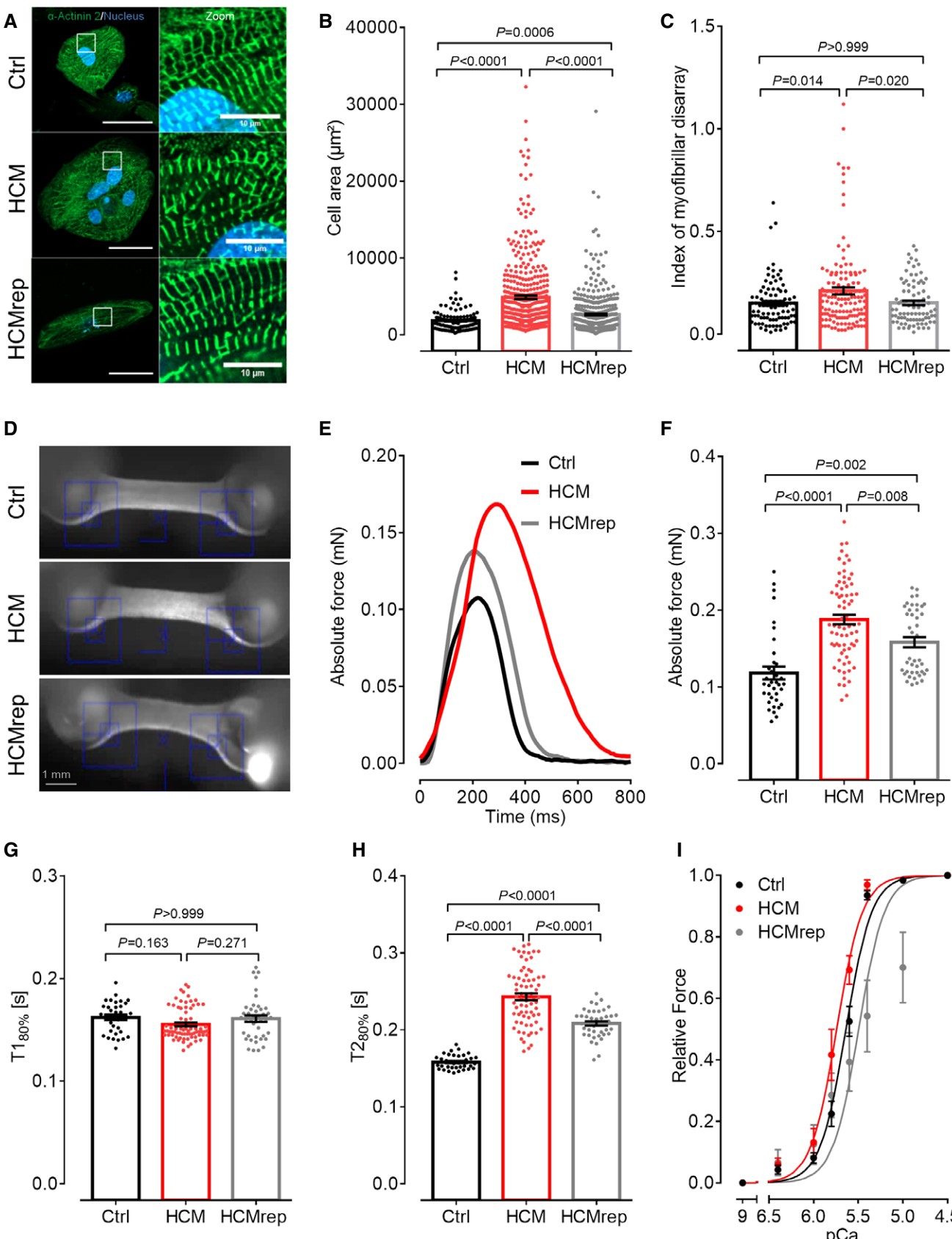

Figure 2.

**Figure 2. Disease modeling with 2D- and 3D-cultured hiPSC-derived CMs.**

A   Representative immunofluorescence images of the Ctrl, the HCM, and isogenic control HCMrep. HiPSC-CMs were stained after 30 days in 2D culture with an antibody against α-actinin 2 and Hoechst for nuclei staining (scale: 50 μm) and with higher magnification (scale: 10 μm).

B   Quantification of cell area analyzed with Fiji software [*n* = number of analyzed cells/wells/differentiations: Ctrl (*n* = 112/11/3); HCM (*n* = 424/7/4); HCMrep (*n* = 443/5/3)].

C   Blinded analysis of myofibrillar disarray using high-resolution pictures [*n* = number of analyzed sarcomeres/cells/differentiations: Ctrl (*n* = 90/30/3); HCM (*n* = 132/44/2); HCMrep (*n* = 86/29/2)].

D   Representative EHTs from each cell line.

E   Representative single traces of EHTs in 1.8 mM Ca$^{2+}$ Tyrode's solution under 1-Hz pacing at 37°C.

F–H   Functional parameters of force (F), time to peak 80% (T1$_{80\%}$, G), and time from peak to baseline 80% (T2$_{80\%}$, H) were measured under paced conditions at 1 Hz in 1.8 mM Ca$^{2+}$ Tyrode's solution at 37°C [*n* = number of measurements/EHTs/differentiations: Ctrl (*n* = 36/25/4); HCM (*n* = 64/60/6); HCMrep (*n* = 41/29/4)].

I   Force–pCa relationships were performed with the Aurora 1400A system in skinned EHT muscle strips from respective cell lines [*n* = number of measured and skinned EHT strips/number of EHTs/number of batches: Ctrl (*n* = 14/7/3); HCM (*n* = 10/4/2); HCMrep (*n* = 7/7/2)]. Concentration–response curves were fitted to the data points, and force–pCa relationship comparison was done by using extra sum-of-squares *F*-test.

Data information: Data are expressed as mean ± SEM, one-way ANOVA with Bonferroni's post-test.

HCM-EHTs than in Ctrl- and HCMrep-EHTs (Fig 3A and B; Table 2). Longer APD in HCM was consistent over the whole range of physiologically relevant frequencies between 1 and 3 Hz (Fig 3C).

Prolongation of APs might be explained by an increased L-type Ca$^{2+}$ channel (LTCC) current (I$_{Ca,L}$) in HCM. Accordingly, we measured I$_{Ca,L}$ by patch clamp technique in isolated hiPSC-CM from a confluent 2D culture. The I$_{Ca,L}$ density, expressed as nifedipine-sensitive inward current, was higher in HCM than in Ctrl and HCMrep hiPSC-CMs, respectively (Fig 3D and E). We used an established computer model for AP and force in hiPSC-CM (Lemoine *et al*, 2018) in order to estimate whether the increased I$_{Ca,L}$ could explain the observed APD prolongation. Indeed, increase of I$_{Ca,L}$ by 50% showed similar APD prolongation, delayed relaxation, and enhanced force (Fig 3F), supporting the experimental data (Fig 2E and F). The observed increase in I$_{Ca,L}$ might thus contribute to the electro-mechanical phenotype in this *ACTN2*-associated HCM.

It has been shown previously that α-actinin 2 interacts with ion channels and contributes to their modulation, such as the LTCC complex (Maruoka *et al*, 2000; Lu *et al*, 2007; Eden *et al*, 2016; Hong & Shaw, 2017). In particular, by binding to the IQ segment of the Ca$_v$ α1.2, pore unit of the LTCC, α-actinin 2 modulates LTCC density and function at the plasma membrane (Sadeghi *et al*, 2002; Hall *et al*, 2013; Tseng *et al*, 2017). Since we found increased I$_{Ca,L}$ in HCM hiPSC-CMs and EHTs compared to Ctrl (Fig 3D and E), we tested whether the mutation in α-actinin 2 might affect the interaction affinity between α-actinin 2 and Ca$_v$α1.2. By a bioluminescence resonance energy transfer (BRET) assay performed in live cardiac muscle-like HL-1 cells, we observed a similar binding affinity of Ca$_v$α1.2 to WT and mutant α-actinin 2 (Appendix Fig S4). However, while this

binding affinity of Ca$_v$α1.2 to WT α-actinin 2 strongly increased with the co-transfection of Ca$_v$β2, the LTCC accessory subunit and chaperone of the LTCC pore unit, it did not in case of mutant α-actinin 2. Thus, the HCM mutation decreased the interaction of α-actinin 2 with the LTCC complex, possibly affecting the activity of the channel. This may explain the electro-mechanical phenotype of the investigated HCM-affected family, characterized by higher I$_{Ca,L}$ density.

## Diltiazem intervention ameliorates the disease phenotype in HCM-EHTs and the QT prolongation in HCM-affected family members

To follow the above reasoning, we tested whether the LTCC blocker diltiazem could reverse the contractile and electrophysiological phenotype of HCM-EHTs. Treatment of EHTs with 3 μM diltiazem for 15 min reduced peak force to a significantly greater extent in HCM- (−29%; Fig 4A and C; Movies EV1 and EV2) than in HCMrep-EHTs (−11%; Fig 4B and C; Movies EV3 and EV4). Similarly, diltiazem shortened relaxation to a greater extent in HCM- (−24.2%; Fig 4A and D) than in HCMrep-EHTs (−11.7%; Fig 4B and D). In addition, diltiazem intervention led to a more pronounced AP shortening in HCM- (Fig 4E, G and H) than in HCMrep-EHTs (Fig 4F, G and H). These results demonstrate that HCM-EHTs are more susceptible to I$_{Ca,L}$ block, supporting the hypothesis that increased I$_{Ca,L}$ is an important contributor to the contractile and electrophysiological phenotype in this case of *ACTN2*-associated HCM.

In line with this information, we found that corrected QT intervals (QT$_c$) were in fact prolonged in affected versus non-affected

**Table 2. Action potential parameters in 3D-cultured hiPSC-derived EHT.**

| | Ctr (*n* = 20) | HCM (*n* = 13) | HCMrep (*n* = 14) | *P* values | | |
| --- | --- | --- | --- | --- | --- | --- |
| | | | | Ctrl versus HCM | Ctrl versus HCMrep | HCM versus HCMrep |
| TOP (mV) | −76.1 ± 0.8 | −78.9 ± 1.6 | −75.7 ± 0.8 | 0.181 | 0.683 | 0.062 |
| APA (mV) | 100.9 ± 1.5 | 116.0 ± 2.7 | 106.2 ± 1.4 | 0.003 | 0.962 | 0.003 |
| Vmax (V/s) | 217.2 ± 16.7 | 243.7 ± 39.9 | 208.5 ± 19.1 | 0.730 | 0.965 | 0.623 |
| APD$_{50}$ (ms) | 154.9 ± 12.9 | 298.9 ± 20.5 | 176.3 ± 14.7 | < 0.001 | 0.585 | < 0.001 |
| APD$_{90}$ (ms) | 242.2 ± 12.6 | 364 ± 21.0 | 256.0 ± 17.4 | < 0.001 | 0.814 | < 0.001 |

Summary of results for take-off potential (TOP), action potential amplitude (APA), maximum upstroke velocity (V$_{max}$), AP duration at 50 and 90% repolarization (APD$_{50/90}$; *n* = number of EHTs). *P* values for multiple comparison with one-way ANOVA and Bonferroni's post-test.

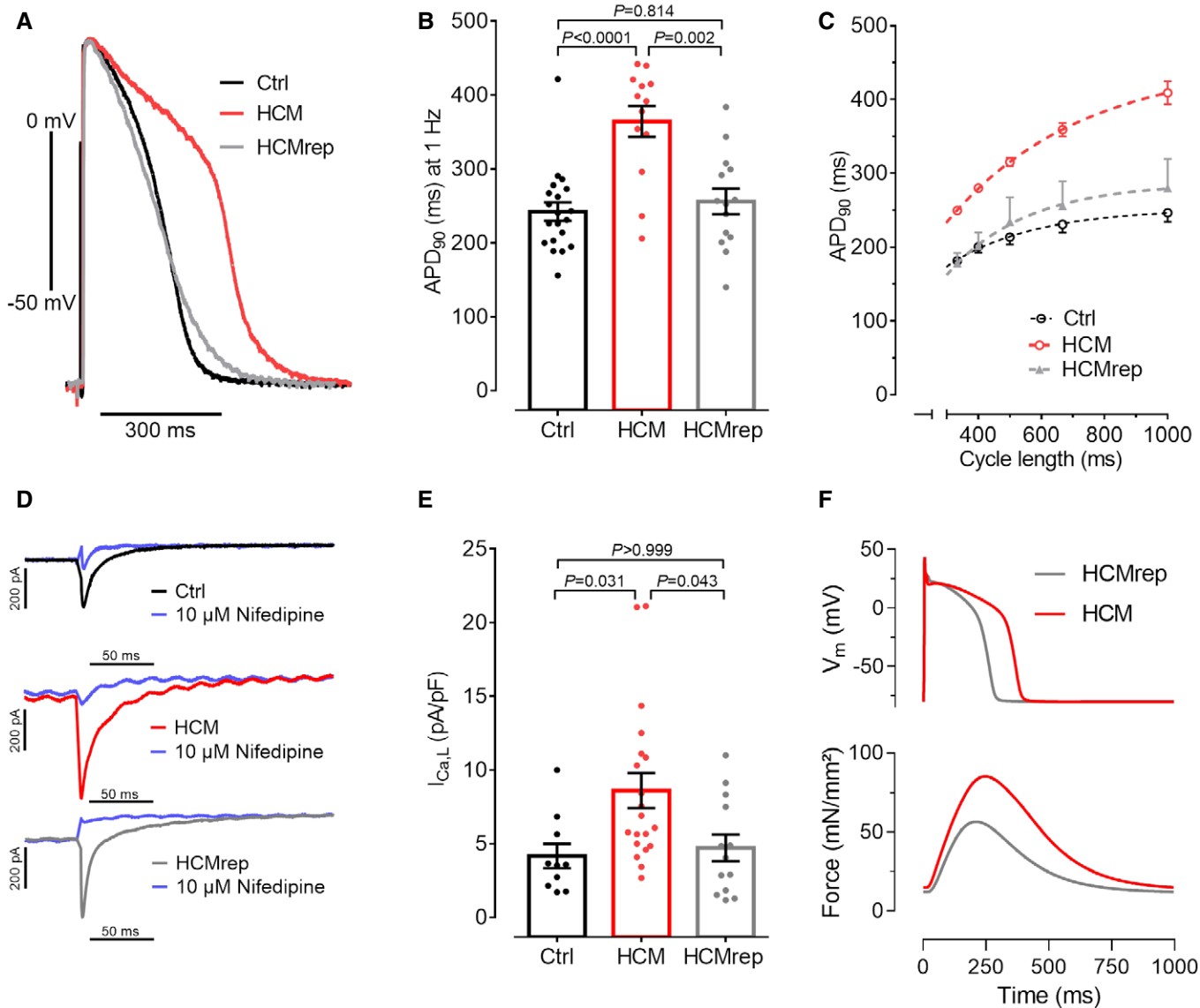

**Figure 3. Electrophysiological characterization with 3D- and 2D-cultured hiPSC-derived CMs.**

A    Representative action potentials (APs) measured with sharp microelectrode in EHTs at 1-Hz pacing.

B    Quantification of AP duration at 90% repolarization (APD$_{90}$) of Ctrl, HCM, and HCMrep [$n$ = number of EHTs/differentiations; Ctrl (20/8), HCM (13/6), HCMrep (14/5)].

C    Cycle length dependence of APD$_{90}$ of Ctrl ($n$ = 7/3), HCM ($n$ = 5/2), and HCMrep ($n$ = 3/1).

D, E   Representative traces (D) and quantification (E) of single-cell patch clamp recordings of L-type calcium currents [$n$ = number of isolated cardiomyocytes/differentiations; Ctrl (10/3), HCM (20/3), HCMrep (13/3)].

F    Simulated effect of 1.5-fold I$_{Ca,L}$ density in *in silico* model of human hiPSC-CM AP and Force.

Data information: Data are expressed as mean ± SEM, one-way ANOVA with Bonferroni's post-test.

family members (Table 1, Appendix Fig S5). This information had not been given special consideration in clinical routine. Since patient II.4 underwent catheter ablation of atypical atrial flutter, we took the opportunity to record monophasic action potentials (MAPs) in the right ventricle (Fig 5A–C) as previously published (Shimizu *et al*, 1991). MAP duration at 90% of repolarization on the right ventricular septum was similar to a cohort with long QT (LQT) syndrome, but longer than in a control group as previously published (Fig 5D; Shimizu *et al*, 1991). In order to exclude a

blended phenotype, we tested the index patient II.4 for mutations in the LQTS genes *CACNA1C, CALM1, CAV3, KCNE1, KCNE2, KCNH2, KCNJ2, KCNJ5, KCNQ1, SCN4B, SCN5A,* and *SNTA1* but could not identify a causative mutation (data not shown).

Based on our *in vitro* findings, we considered clinical implementation of diltiazem to ameliorate the electro-mechanical phenotype of HCM-affected family members. At this time, the index patient II.4 was still receiving antiarrhythmic treatment with amiodarone and bisoprolol because of recurrent atypical atrial flutter. Since we

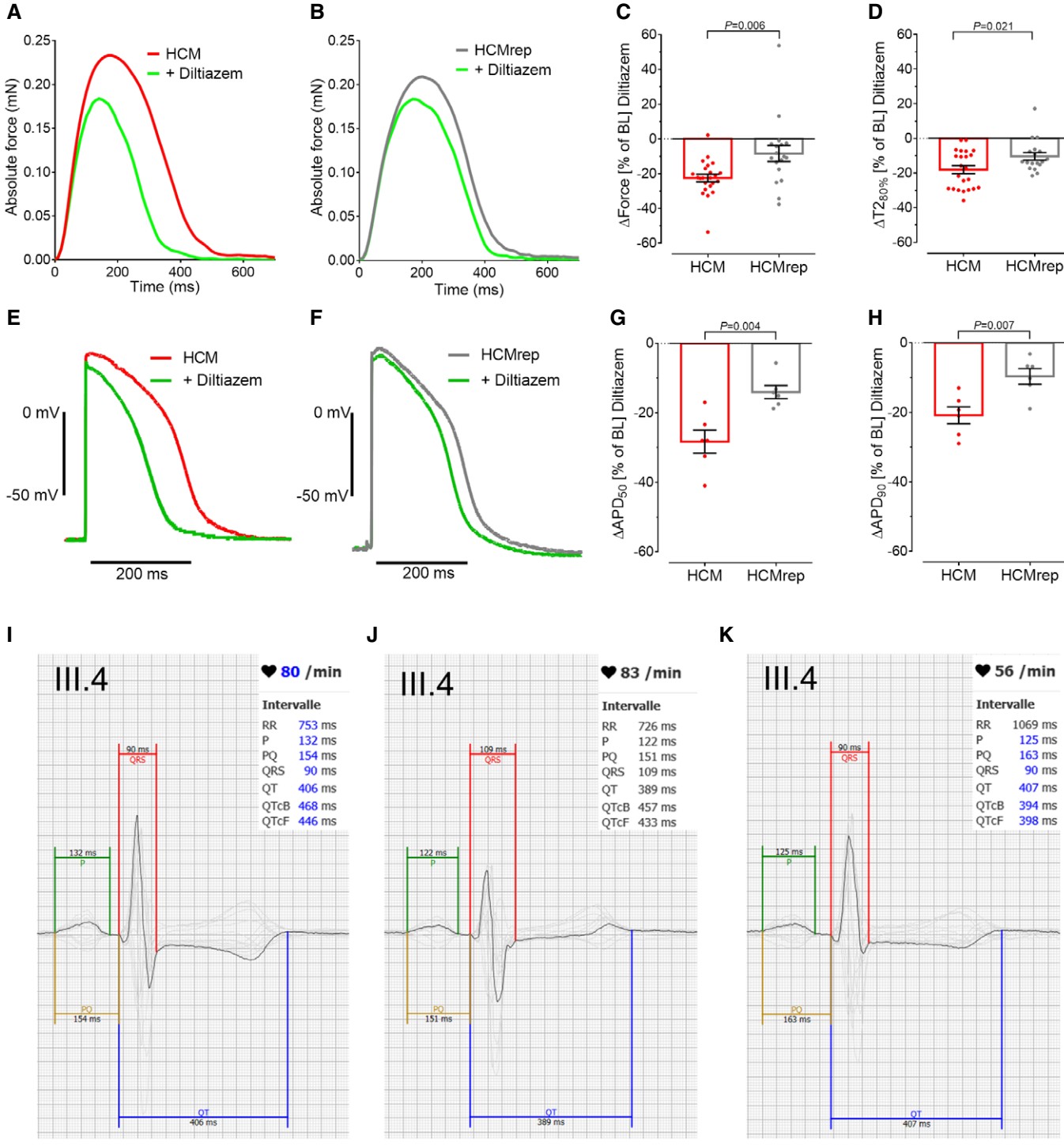

**Figure 4. Diltiazem intervention in 3D-cultured hiPSC-derived EHT.**

A, B    Representative force traces at 1-Hz pacing under baseline condition and with 3 μM diltiazem in HCM- (A) and HCMrep-EHTs (B).

C, D    Quantification of the diltiazem effect in percent to baseline (BL) levels for force (C) and time to baseline 80% [T2$_{80\%}$, D; n = number of EHTs/differentiations; HCM (n = 23/2); HCMrep (n = 18/2)].

E, F    Representative action potentials (AP) paced at 1 Hz under BL condition (red for HCM, gray for HCMrep) and with 3 μM diltiazem (green) in HCM- (E) and HCMrep-EHTs (F).

G, H    Quantification of AP duration at 50% (G) and 90% (H) of repolarization in HCM (n = 6/1) and HCMrep (n = 6/1).

I–K    Representative 12-lead surface ECG with averaged signal and overlay of all leads (in gray, apart from V5 in black) of patient III.4 with different antihypertensive medication: (I) ramipril 5 mg, (J) ramipril 10 mg + 5 mg amlodipine, (K) ramipril 5 mg + diltiazem 180 mg for 1 month (QTcB: QT interval with Bazett correction; QTcF: with Fredericia correction).

Data information: Data are expressed as mean ± SEM, unpaired Student's t-test.

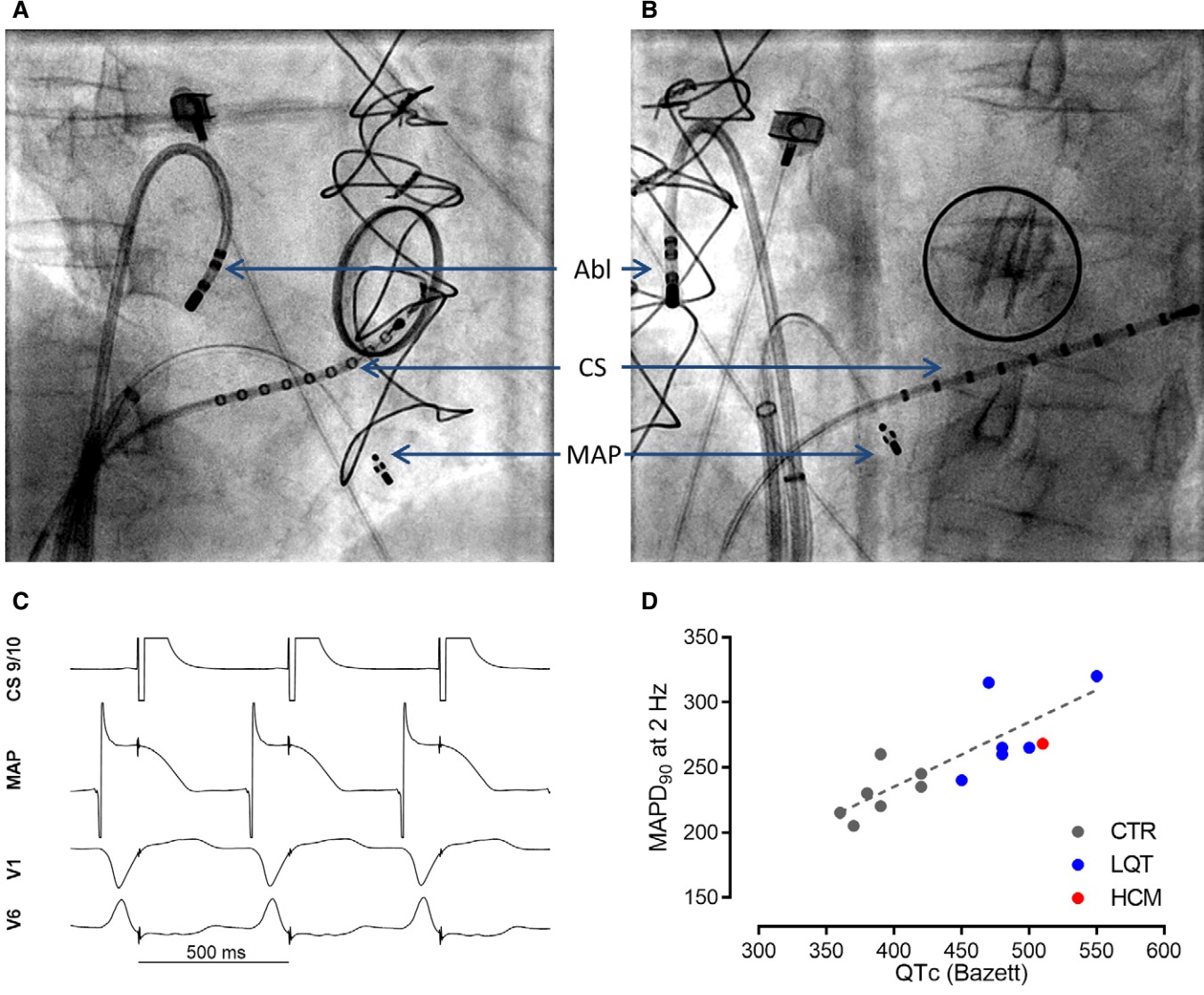

**Figure 5. Monophasic action potential from HCM-affected family member II.4.**

A, B  Fluoroscopy during the recording of monophasic action potentials in right anterior oblique projection (A) and left anterior oblique projection (B) at 30°. Atrial stimulation was performed by a catheter in coronary sinus (CS) and a second catheter for the recording of monophasic action potentials (MAPs) at the septum of the right ventricle, and the ablation catheter (Abl) was not used for the recording.

C  Representative MAPs traces.

D  Comparison of MAPD$_{90}$ and QTc (Bazett) in the index patients II.4 to a control and long QT (LQT) cohort taken from Shimizu et al (1991).

would not like to combine diltiazem with amiodarone and bisopro- lol, we contacted patient III.4, who was receiving antihypertensive treatment with ramipril plus amlodipine, and patient II.5, who was receiving candesartan only. In case of patient III.4, we replaced amlodipine (a LTCC blocker with preferential vasodilatory effects) with diltiazem (a LTCC blocker with preferential cardiac action) (Lee & Tsien, 1983). At a standard dose (90 mg diltiazem, twice daily), QTc was reduced from 460 to 387 ms (Fig 4I–K). Reduced QTc was confirmed by 12-lead ECGs over 24 h (averaged QTc $\pm$ 95% confidence interval): 446 $\pm$ 29 ms (5 mg ramipril, 5 mg amlodipine) to 418 $\pm$ 35 ms (5 mg ramipril, diltiazem 180 mg). In case of patient II.5, we added diltiazem (90 mg twice daily) and reduced candesartan from 8 to 4 mg. In consequence,

QTc in 12-lead ECG was reduced from 477 to 439 ms. These clinical data support our *in vitro* findings that increased I$_{Ca,L}$ contributes to the electrophysiological phenotype and that its reduction by an LTCC blocker is efficient.

## Discussion

In this study, we applied hiPSCs and CRISPR/Cas9 gene editing to model a novel *ACTN2* missense mutation in an HCM-affected family and showed that HCM hiPSC-CMs recapitulated known hallmarks of HCM such as myofibrillar disarray, cell hypertrophy, increased myofilament Ca$^{2+}$ sensitivity, hypercontractility, and prolonged

relaxation. In addition, the *ACTN2* mutation resulted in AP prolongation and abnormal high $I_{Ca,L}$ density. Given the availability of a clinically approved drug that reversed the contractile and electrophysiological phenotype *in vitro* (diltiazem), we could directly translate our findings to patient care and showed that diltiazem at standard dose reverted the LQT phenotype in the son and sister of the index patient. This study shows that patient-specific hiPSC-CMs can be used not only to describe disease-related phenotypes, but also to individualize patient care for this specific *ACTN2* mutation.

The level of evidence for *ACTN2* being a causative gene in HCM is not strong as compared to *MYBPC3* and *MYH7* genes (Walsh *et al*, 2017). However, we provide several lines of evidence for this novel *ACTN2* mutation to be HCM-causing: (i) It was segregated in three HCM-affected members of the family, (ii) it was not found in the gnomAD browser, assembling > 130,000 exome or whole-genome sequences from unrelated control individuals, and (iii) the patient-derived hiPSC-CMs developed a phenotype, which was corrected in the isogenic control. Our disease modeling approach revealed HCM phenotypes in 2D- and 3D-cultured hiPSC-CMs that were also reported previously in hiPSC-CMs and animal models, such as myofibrillar disarray and CM hypertrophy (Lan *et al*, 2013; Dambrot *et al*, 2014; Tanaka *et al*, 2014; Ojala *et al*, 2016; Prondzynski *et al*, 2017), higher contraction force, longer relaxation time (Wijnker *et al*, 2016), and higher myofilament $Ca^{2+}$ sensitivity (Fraysse *et al*, 2012; Eschenhagen & Carrier, 2019). The fact that all parameters were significantly different between the HCM and the isogenic control line carrying the same genetic background provides evidence for the mutation to be causative. However, significant differences in cell size, force, and relaxation time were also observed between HCMrep-CMs and Ctrl-CMs. Variability between control cell lines has been reported in several studies. For example, the response to diltiazem and other compounds varied between several control hiPSC-CM lines (Lam *et al*, 2019). We also previously found significantly different APD90 in three different control hiPSC-cardiomyocyte cell lines (Lemoine *et al*, 2018). More recently, we systematically analyzed in a blinded manner 36 control cell lines and found a standard deviation of the mean of 20–35% for several parameters of EHT contractile function (*Eschenhagen, personal communication*). Taken together, these studies underline the need of isogenic controls for personalized medicine.

The absence of a difference in α-actinin 2 protein levels between HCM and control models combined with allelic balance of WT and mutant *ACTN2* mRNA transcripts suggests that the molecular mechanism of this *ACTN2* mutation is not haploinsufficiency but rather a gain-of-function mechanism with translation of mutant mRNA to an α-actinin 2 protein with altered functional properties. We provide evidence for a defective $Ca_v\beta2$-dependent interaction of mutant α-actinin 2 with the LTCC $Ca_v\alpha1.2$ pore-forming unit, suggesting that WT α-actinin 2 plays a relevant role in regulating LTCC assembly and function and that the channel activity is disturbed by the mutation. The physical interaction of α-actinin 2 with $Ca_v\alpha1.2$ has previously been shown in human heart tissue by co-immunoprecipitation (Lu *et al*, 2007) and is further dependent on the LTCC $Ca_v\beta2$ auxiliary subunit as shown by our data. Since the HCM mutation is localized in the actin-binding domain of α-actinin 2 and not within the LTCC interacting region, we propose that the LTCC complex is correctly placed at the plasma membrane, but due to defective interaction with mutant α-actinin 2, functional properties might be altered (Hall *et al*, 2013). Further studies will be dedicated to the unraveling of underlying mechanisms.

In order to translate *in vitro* findings to patient care, we evaluated how the electro-mechanical phenotype could be pharmacologically modulated. HCM-EHTs showed a significantly longer AP duration with a prolonged plateau phase, which could be caused by the higher $I_{Ca,L}$ density. Indeed, diltiazem reduced force amplitude, relaxation, and APD to a greater extent in HCM- than in HCMrep-EHTs. Similarly, restoration of diastolic function with diltiazem has been recently reported in hiPSC-CMs carrying HCM mutations in *MYBPC3*, *MYH7*, or *TNNT2* (Wu *et al*, 2019). Prolonged APs in EHTs were in line with prolonged QTc intervals in all family members carrying the *ACTN2* mutation and with prolonged MAPs at the right ventricular septum of the index patient II.4. Therefore, we started treatment with diltiazem in patients III.4 and II.5 as an antihypertensive treatment. All clinically established LTCC blockers have both vasodilatory and cardiodepressant effects, only the ratio between the two principle actions differ (Eschenhagen, 2018). Diltiazem, in contrast to amlodipine, has relevant negative chronotropic, inotropic, and dromotropic effects at clinically used doses. The observation that the switch from amlodipine to diltiazem reduced the prolonged QTc interval in patient III.4 supports the conclusion that the increase in $I_{Ca,L}$ plays a pathophysiological relevant role in this patient. In healthy controls, diltiazem therapy revealed no QTc-shortening effect previously (Ritterman *et al*, 1982), supporting our experimental results that control hiPSC-CMs showed less effect upon diltiazem. A previous publication revealed a causative *ACTN2* mutation for HCM with a pro-arrhythmic phenotype (Bagnall *et al*, 2014). Although patients III.4 and II.5 have not suffered from excessive ventricular extrasystole or ventricular tachycardia yet, normalized QTc interval by $I_{Ca,L}$ inhibition might reduce the risk of sudden cardiac death by preventing cellular $Ca^{2+}$-overload and triggered arrhythmia as shown in a mouse model of HCM (Westermann *et al*, 2006). In a large HCM cohort, the mean QTc was only mildly prolonged (440 ms versus 425 ms in controls) (Johnson *et al*, 2011), but in a subgroup of genetically identified HCM patients with mild phenotype and symptoms, 9% showed a QTc > 480 ms, suggesting that impaired repolarization may be not an uncommon feature of the disease. Disease modeling of several different mutations in hiPSC-CMs showed mostly abnormal $Ca^{2+}$ handling but shortened, unchanged, or prolonged APD, paralleling the clinical heterogeneity (reviewed in Eschenhagen & Carrier, 2019). Examination of the patient's further clinical course will show whether normalization of the QTc interval as a surrogate of clinical improvement will be meaningful and if diltiazem treatment will positively affect other HCM parameters in this patient.

Taken together, this "case study" revealed a novel, rare HCM mutation associated with a contractile and electrophysiological phenotype in hiPSC-CMs/-EHTs that guided clinical therapy in two HCM patients. We believe that this study may pave the way for more personalized approaches for less evident HCM gene or variants of uncertain significance. While pharmacological intervention with diltiazem acutely ameliorated the prolonged QTc, longer-term follow-up is necessary to see whether diltiazem also affects progression of structural aspects of the disease. The interesting fact that the prolonged QTc did not find attention during clinical routine but was revealed after *in vitro* studies supports the view for more personalized medicine. Given the relevant frequency of QTc prolongation in

HCM and the possibility that other HCM mutations as well as myocardial disarray in a more general sense may affect LTCC activity (Viola & Hool, 2017), future work should be directed toward answering whether the beneficial effects of diltiazem treatment observed in our study and in a previously published pilot study (Ho et al, 2015) may in fact relate to a general mechanism of HCM pathology or might be specific due to altered interaction between α-actinin 2 and the LTCC, as indicated by our BRET analysis and recently identified (Tseng et al, 2017).

# Materials and Methods

### Human material

The HCM patient carrying the ACTN2 (c.740C>T; dbSNP ID: rs755492182) mutation was recruited in the outpatient clinic at the University Heart Center Hamburg and provided written informed consent for genetic analysis and the use of fibroblasts. Cardiac tissue of the patient was obtained during septal myectomy. Control donor tissues were from non-failing human heart tissues not suitable for transplantation or from donors that did not die from cardiac disease but of another cause. All materials were taken with informed consent of the patients and donors. The study was reviewed and approved by the Ethical Committee of the Ärztekammer Hamburg (PV3501), and the index patient gave written informed consent. This study is in accordance with the Code of Ethics of the World Medical Association (Declaration of Helsinki).

### Identification of HCM mutation

Genomic DNA was obtained from peripheral blood by standard procedures, and mutations in 19 known or suspected HCM genes were detected. Gene target enrichment was undertaken using HaloPlex Target Enrichment System (Agilent) and sequencing carried out on an Illumina MiSeq Desktop Sequencer. Sequence data were analyzed using a custom-designed bioinformatic pipeline. The presence of the mutation was confirmed by Sanger sequencing in DNA derived from blood leukocytes. Segregation analysis of the mutation was also performed by Sanger sequencing according to standard procedures (details are available upon request). For in silico pathogenicity prediction, the programs CADD (Combined Annotation Dependent Depletion) (Kircher et al, 2014), REVEL (Rare exome variant ensemble learner) (Ioannidis et al, 2016), and M-CAP (Mendelian Clinically Applicable Pathogenicity) (Jagadeesh et al, 2016) were used. Pathogenicity thresholds were ≥ 20 (CADD), ≥ 0.3 (REVEL), and ≥ 0.025 (M-CAP).

### Electrocardiography

Electrocardiographic analysis was performed by using a 12-lead ECG using an automated analyses program (Schiller Cardiovit AT-10 plus Felsberg, Germany). All QT and RR intervals were re-analyzed manually and averaged from by three reviewers independently and blinded for genotype and drug intervention. Mean QT intervals (Table 1) deviate slightly from representative ECG measurements from one reviewer (Fig 4I–K; Appendix Fig S5). QT intervals were measured in lead II or $V_5$ when possible and under consideration of

all leads. QTc values were calculated using the Bazett (QTcB = QT/$(RR^{(1/2)})$ and Fredericia (QTcF = QT/$(RR^{(1/3)})$) correction (Rautaharju et al, 2009; Vandenberk et al, 2016). For the index patient, we measured QT intervals in ECGs recorded before the surgical myectomy because of a post-operative total left bundle branch block (Appendix Fig S5A) which does not allow a valid QT interpretation.

### Measurement of monophasic action potentials

Written informed consent was obtained before the study from the index patient. Bisoprolol had been discontinued for 48 h before the procedure. MAPs were measured after the ablation procedure in stable sinus rhythm. MAPs were recorded at different positions on right ventricular septum by a 4-pole catheter (EasyMAP, MedFact, Lörrach, Germany) during constant right atrial pacing of 2 Hz as described before (Shimizu et al, 1991). Signals of MAPs were recorded at 1 kHz and filtered at a frequency of 0.05–500 Hz (BARD LabSystem, Bard Electrophysiology, Murray Hill, MA).

### Next-generation sequencing for LQT syndrome

After library preparation by the TruSight™ Rapid Capture Kit and targeted enrichment by the TruSight™ Cardio Kit (Illumina, San Diego, CA), cluster generation and sequencing was performed on a MiSeq flow cell with the MiSeq Reagent Kit v.2 on a MiSeq platform (Illumina) according to the manufacturer's instructions. Data analysis was done by Sequence Pilot module SeqNext (JSI Medical Systems, Kippenheim, Germany) with the reference sequences NM_000719.6 (CACNA1C), NM_006888.4 (CALM1), NM_033337.2 (CAV3), NM_000219.5 (KCNE1), NM_172201.1 (KCNE2), NM_000238.3 (KCNH2), NM_000891.2 (KCNJ2), NM_000890.3 (KCNJ5), NM_000218.2 (KCNQ1), NM_174934.3 (SCN4B), NM_198056.2 (SCN5A), and NM_003098.2 (SNTA1). According to the manufacturer's instructions, Multiplex ligation-dependent probe amplification (MLPA®) was performed with the kit P114-B2 (MRC Holland) for the genes KCNE1, KCNE2, KCNH2, and KCNQ1. Data were analyzed by the software Sequence Pilot module MLPA (JSI Medical Systems) after capillary electrophoresis on an automated capillary sequencer (ABI 3500; Applied Biosystems).

### Generation and culture of hiPSC-derived cardiomyocytes in 2D and in EHT format

A skin biopsy was taken, washed in PBS, minced, and placed in a 6-well plate in fibroblast medium [DMEM with 10% FBS (PAA), 2 mM ʟ-glutamine and 0.5% penicillin and streptomycin (all Life Technologies)]. Dermal fibroblasts growing out of the explants were collected for passaging or cryopreservation and used for subsequent reprogramming at passage 5. The reprogramming was performed according to previously published protocols with retroviruses encoding the human transcription factors OCT3/4, SOX2, KLF4, and L-MYC (Takahashi et al, 2007a,b; Ohnuki et al, 2009). Fibroblast from the HCM patient with the ACTN2 (c.740C>T) mutation was reprogrammed using the CytoTune-iPS Sendai Reprogramming Kit (Product No. A1377801, Life Technologies). CM differentiation from hiPSCs was performed following a three-step protocol with generation of embryoid bodies (EBs) in spinner flasks and FACS analysis, using a cardiac troponin T FITC-labeled antibody (1:10, Miltenyi

Biotec) for CMs quantification (Breckwoldt *et al*, 2017; Appendix Fig S6). Dissociation was done with collagenase 2 (200 units/ml, Worthington, LS004176), and beating hiPSC-CMs were plated on Geltrex-coated (1:100, Gibco) 96-well or 12-well plates at a density of 2,500–5,000 cells/well or 440,000 cells/well, respectively. HiPSC-CMs were maintained in culture for 30 days at 37°C in 7% $CO_2$ and atmospheric $O_2$ (21%) prior to further analysis. Furthermore, single-cell suspensions of hiPSC-CM were subjected to the 3D format of EHTs, which were generated in a 24-well format (800,000 hiPSC-CM/EHT in a fibrin matrix (total volume 100 μl) consisting of 10 μl/100 μl Matrigel (BD Biosciences, 256235), 5 mg/ml bovine fibrinogen [200 mg/ml in NaCl 0.9% (Sigma, F4753) plus 0.5 μg/mg aprotinin (Sigma, A1153), 2× DMEM, 10 μmol/l Y-27632, and 3 U/ml thrombin (Biopur, BP11101104), as described previously (Mannhardt *et al*, 2016; Breckwoldt *et al*, 2017)]. EHTs were maintained in culture at 37°C in 7% $CO_2$ and 40% $O_2$, subjected to further analysis from day 26 on. Medium exchange for 2D and EHT cultures was performed every Monday, Wednesday, and Friday using DMEM [10% heat-inactivated FCS (Gibco), 0.1% insulin (Sigma-Aldrich), 0.5% penicillin/streptomycin (Gibco), 0.1% Aprotinin (Sigma)]. Silicone racks (C001), Teflon spacers (C002), and EHT analysis equipment (A001) were purchased from EHT Technologies GmbH (Hamburg, Germany).

### Generation of the isogenic control cell line HCMrep

For generation of an isogenic control cell line, patient-specific hiPSCs carrying the *ACTN2* (c.740C>T) mutation were used, hereafter named HCM cell line. CRISPR/Cas9 technology was used to repair the patient-specific mutation according to Ran *et al* (2013). In short, Cas9 nickase approach was used for induction of a double-strand break (DSB) in the desired locus. Single-guide RNAs (sgRNAs) were designed using a publically available tool (crispr.mit.edu) and cloned into the pSpCas9n(BB)-2A-GFP plasmid (Addgene). Homology-directed repair (HDR) was supported by supplying an exogenous repair template encoding the WT sequence of *ACTN2* from position g.54,148-54,270 (NG_009081.1; Accession number NCBI). Repair templates were produced by IDT® as single-stranded oligonucleotide donors (ssODNs). Silent mutations were introduced at genomic positions 54,200 and 54,245 within the protospacer adjacent motif (PAM) sequence for prevention of repeated cutting events and verification of successful genome editing. HCM hiPSCs were transfected with the Amaxa Nucleofector™ (Lonza) using the P3 Primary Cell 4D-Nucleofector® X Kit and the pulse code CA 137. Each transfection approach consisted of 800,000 HCM hiPSCs, 40 μM of ssODN, 1,000 ng of each plasmid pSpCas9n(BB)-2A-GFP (Addgene) containing sgRNA A and B. One hour before transfection, HCM hiPSCs were treated with 10 μM ROCK inhibitor Y-27632 (Y; Biorbyt). For transfection, HCM hiPSCs were washed twice with PBS and singularized with accutase (Gibco®; 5 min, 37°C at 5% $CO_2$). Enzymatic dissociation was stopped by adding the same volume of conditioned medium (COM) containing bFGF (30 ng/ml; Peprotech) and 10 μM Y. After counting, 800,000 cells were spun down for 3 min at 150 g and subsequently resuspended in the transfection mix and transferred into electroporation cuvettes (Lonza). After transfection, electroporation cuvettes were incubated for 5 min at 37°C in 5% $CO_2$. 500 μl pre-warmed COM, supplemented with FGF (30 ng/ml) and 10 μM Y, was added to the transfection mix that subsequently was plated onto

Matrigel-coated (1:60; Corning®) 12-well plates. Transfected HCM hiPSCs were maintained for 48 h and prepared for fluorescent activated cell sorting (FACS) by accutase dissociation as described above. FACS was performed using the FACSAria™ III (BD; FACS Core facility UKE Hamburg) collecting GFP-positive cells. GFP-positive cells were centrifuged for 3 min at 150 g, and 2,500 cells were plated into one well of a Matrigel-coated 6-well plate. Cells were maintained by daily medium change until colonies were visible. Picking was carried out using a 200-μl pipette, whereby each clone was transferred into one well of a Matrigel-coated 48-well plate. Clones were maintained until sufficient cell material was generated for cryopreservation and genomic analysis by PCR. If genomic analysis revealed successfully repaired HCM hiPSC clones, these clones were subcloned by plating 1,000 cells into one well of a Matrigel-coated 6-well plate. Subcloned repaired HCM hiPSC clones were again picked and cultured, until characterization could be repeated as described above. Furthermore, these clones were analyzed for off-targets by PCR (Appendix Table S2). Subcloning was introduced to minimize chances of mixed clonal populations. Additionally, PCR fragments of cells containing the modified *ACTN2* locus were subcloned using the CloneJET PCR Cloning Kit (Thermo Fisher Scientific) for discrimination of allele-specific genotypes. Finally, all used cell lines were genotyped by PCR for the *ACTN2* locus and karyotyped by G-banding as reported previously (Breckwoldt *et al*, 2017). An overview of this procedure is depicted in Appendix Fig S1A.

### Genotyping and off-target analysis using polymerase chain reaction

To amplify specific DNA fragments, PCR was conducted according to Mullis (Mullis & Faloona, 1987). The applied PCR program was adapted to the used polymerase and annealing temperatures of the primers (listed in the data sheet supplied with the primers). The elongation time was adjusted to the length of the expected DNA fragment, and the synthesis rate was estimated to be 1 kb/min, unless noted otherwise by the manufacturer. Genomic DNA template was used in an amount of 20–40 ng in a final volume of 20 μl and 50–100 ng in a final volume of 50 μl. All used primers are listed in Appendix Table S3. For genotyping of putative CRISPR/Cas9-modified HCM hiPSCs clones, touchdown PCR (60°C–55°C) was performed using PrimeStar® HS DNA Polymerase in a 50-μl PCR approach, for 35 cycles according to the instructions of the manufacturer's protocol. For off-target analysis of successfully repaired HCM hiPSCs clones, touchdown PCR (60°C–55°C) was performed using AmpliTaq® Gold DNA polymerase in a total volume of 20 μl for 35 cycles according to the instructions of the manufacturer's protocol.

### Analysis of allele-specific mRNA transcripts

Quantification of missense and WT mRNA was done with total RNA of EHTs and of myectomy tissue received from the index patient. RNA (200 ng) was reverse-transcribed to cDNA according to the instructions of the manufacturer's protocol (SuperScript™ III First-Strand Synthesis System, Invitrogen). cDNA was amplified with PrimeStar® HS DNA Polymerase in a 50-μl PCR approach, for 35 cycles according to the instructions of the manufacturer's protocol (Primers, Appendix Table S3). Generated PCR fragments were

purified with the QIAquick PCR Purification Kit (Qiagen) and cloned with the CloneJET PCR Cloning Kit (Thermo Fisher Scientific) for discrimination of allele-specific mRNA transcripts. Single colonies of transformed One Shot® TOP10 E. coli were transferred to overnight cultures [5 ml, lysogeny broth (BD), ampicillin (100 μg/ml; Serva)]. Plasmid extraction (NucleoSpin® Plasmid, Macheray-Nagel) was performed the next day and send for sequencing (Eurofins Genomics).

### RNA isolation and gene expression analysis

Total RNA was extracted from 2D-cultured hiPSC-CMs and -EHTs using TRIzol® Reagent (Life Technologies) following the manufacturer's protocol. RNA extraction from human heart tissue was performed with the SV total RNA isolation Kit (Promega). For RNA sequencing analysis, RNA integrity was analyzed with the RNA 6000 Nano Chip on an Agilent 2100 Bioanalyzer (Agilent Technologies). From total RNA, mRNA was extracted using the NEBNext Poly(A) mRNA Magnetic Isolation module (New England Biolabs) and RNA-Seq libraries were generated using the NEXTFLEX Rapid Directional qRNA-Seq Kit (Bioo Scientific) as per the manufacturer's recommendations. Concentrations of all samples were measured with a Qubit 2.0 Fluorometer (Thermo Fisher Scientific), and fragment lengths distribution of the final libraries was analyzed with the DNA High Sensitivity Chip on an Agilent 2100 Bioanalyzer (Agilent Technologies). All samples were normalized to 2 nM and pooled equimolar. The library pool was sequenced on the NextSeq 500 (Illumina) with $1 \times 75$ bp, with 16.6–23.8 mio. reads per sample. For each sample, the sufficient quality of the raw reads was confirmed by FastQC v0.11.8 (http://www.bioinformatics.babraham.ac.uk/projects/fastqc). Afterward, the reads were aligned to the human reference genome GRCh38 with STAR v2.7.0f (Dobin et al, 2013) and simultaneously counted per gene by employing the quantmode GeneCounts option. Counts are based on the Ensembl annotation release 96. Normalization factors for each sample were estimated with DESeq2 v1.24.0 (Love et al, 2014) and used to generate normalized coverage tracks with bamCoverage from deepTools v3.2.1 (Ramirez et al, 2016). Mutations and expressions were visualized with Gviz v1.28.0 (Hahne & Ivanek, 2016). For expression analysis with the nanoString nCounter® Elements technology, a total amount of 50 ng RNA was hybridized with a customized nanoString Gene Expression CodeSet (Appendix Table S4) and analyzed using the nCounter® Sprint Profiler. mRNA levels were determined with the nSolver™ Data Analysis Software including background subtraction using negative controls and normalization to six or four (2D-cultured hiPSC-CMs and EHTs, indicated in Appendix Fig S2D) housekeeping genes (ABCF1, CLTC, GAPDH, ACTB, PKG1, TUBB; Appendix Table S4) and expressed as fold change over Ctrl for the respective groups.

### Western Blot analysis

Western blot analysis was performed on total crude protein lysates from human tissue, 2D-cultured hiPSC-CMs, and EHTs. Lysates of single samples or pooled samples were separated on 10% acrylamide/bisacrylamide (29:1) gels and transferred by wet electroblotting to nitrocellulose membranes. Membranes were stained with primary antibodies directed against α-actinin 2 (1:1,000, Sigma) and

cardiac troponin T (1:5,000, Abcam). Peroxidase-conjugated secondary antibodies against mouse (1:5,000, Sigma) or against rabbit (1:6,000, Sigma) were used. Signals were revealed with the Clarity Western ECL Substrate (Bio-Rad) and acquired with the ChemiDoc Touch Imaging System (Bio-Rad). Signals were quantified with the Image Lab software (Bio-Rad).

### Immunofluorescence staining of hiPSC-derived cardiomyocytes

HiPSC-CMs were cultured for 30 days in 96-well plates (μclear®, Greiner) and subsequently prepared for immunofluorescent analysis, as described previously (Prondzynski et al, 2017). Staining for α-actinin 2 was performed by a primary antibody (1:800, Sigma) and a secondary anti-mouse Alexa Fluor® 488 antibody (1:800, Life Technologies). The nucleus was stained with Hoechst 33342 (1:2,500, Thermo Fisher Scientific), and images were obtained by confocal microscopy using the Zeiss LSM 800 confocal microscope.

### Morphological analysis of 2D-cultured hiPSC-derived cardiomyocytes

Quantification of myofibrillar disarray and cell area was done by using Fiji software (ImageJ), whereby cell area was assessed as described previously (Prondzynski et al, 2017). Myofibrillar disarray was measured by drawing randomly three regions of interests (ROI) in single hiPSC-CMs cultured in 96-well plates. Two lines were drawn with a length of approximately 10 μm for each ROI, spanning ~6 sarcomere intervals, perpendicular to the sarcomere alignment in a distance of 5–10 μm. Myofibrillar disarray was quantified by dividing the parallel intersections of the sarcomere with the two lines and quantifying the standard deviation of each crossed intersection, resulting in the index of myofibrillar disarray. Analysis of myofibrillar disarray was performed in a blinded fashion.

### Analysis of contractile force in EHTs

Contractile force was analyzed as previously described (Mannhardt et al, 2016). In short, > 26-day-old EHTs were incubated in modified Tyrode's solution at least 2 h before starting an experiment (in mM: NaCl 120, KCl 5.4, $MgCl_2$ 1, $CaCl_2$ 1.8, $NaH_2PO_4$ 0.4, $NaHCO_3$ 22.6, glucose 5, $Na_2EDTA$ 0.05, and HEPES 25) and incubated at 37°C in 7% $CO_2$ and 40% $O_2$. The 24-well plate carrying the EHTs was placed inside a transparent chamber to maintain homeostatic temperature (37°C) and $CO_2$ (5%). Automated video-optical recordings of silicon post deflection were enabled by a video camera placed above the chamber. Tracking of EHT movement and determination of contractile force was analyzed based on the known mechanical properties of the silicon using customized software (CTMV, Pforzheim, Germany). EHTs were electrically paced for experimental analysis (1 V, 2 Hz, impulse duration 4 ms). The contraction peaks were analyzed for force amplitude, contraction time ($T1_{80\%}$), and relaxation time ($T2_{80\%}$) at 80% of peak height.

### Myofilament Ca²⁺ sensitivity measurements of human tissue and EHTs

Cardiac muscle strips of $2.69 \pm 0.82$ mm length, $0.32 \pm 0.08$ mm width, and $0.20 \pm 0.09$ mm² cross-sectional area (CSA), calculated

by $2\pi r^2$ assuming a circular shape, were isolated from human cardiac tissues and EHTs. For contractile measurements, strips were permeabilized in a pCa9 EGTA-buffer containing 1% Triton X-100 at 4°C for 18 as described previously (Kooij *et al*, 2010; Stoehr *et al*, 2014; Friedrich *et al*, 2016). The next day strips were either used directly for measurements or stored at −20°C in a 50% glycerol/relaxing solution with protease inhibitors (EDTA-free, complete tablets, mini, Roche). Measurement of contractile function was performed using a fiber test system (1400A; Aurora Scientific) as reported previously (Friedrich *et al*, 2016; Stucker *et al*, 2017). A Hill equation was fitted to the data points (Hill *et al*, 1980) to estimate $pCa_{50}$ as the free $Ca^{2+}$ concentration yielding 50% of the maximal force and nH representing the Hill coefficient. The $pCa_{50}$ represents the parameter of myofilament $Ca^{2+}$ sensitivity.

### Action potential measurements of EHTs

APs were recorded as described previously (Wettwer *et al*, 2013; Lemoine *et al*, 2017) with standard sharp microelectrodes in intact EHTs, field-stimulated at 1 Hz (50% above the stimulation threshold). Tissues were continuously superfused with Tyrode's solution containing (in mM): NaCl 127, KCl 5.4, $MgCl_2$ 1.05, $CaCl_2$ 1.8, glucose 10, $NaHCO_3$ 22, $NaHPO_4$ 0.42, equilibrated with $O_2$-$CO_2$ [95:5] at 36°C, pH 7.4. Sharp microelectrodes were made from filamented borosilicate glass capillaries (external diameter 1.5 mm, internal diameter 0.87 mm; HILG1103227; Hilgenberg, Malsfeld, Germany) by a DPZ-Universal puller (Zeitz Instruments, Munich, Germany). The pipettes had a resistance of 20–50 MΩ when filled with 3 M KCl. Pipettes were mounted on a hydraulic micromanipulator (Narishige MO-203) for gentle impalement in the tissue. Signals were amplified by BA-1s npi amplifier (npi electronic, Tamm, Germany). EHTs were detached from the silicon posts and pinned down without any stretching. To measure rate adaptation, we allowed preparations to equilibrate for at least 300 beats. To measure diltiazem effect, we allowed preparations to equilibrate for at least 10 min after drug exposure. APs were recorded and analyzed using the Lab Chart software (ADInstruments, Spechbach, Germany).

### Whole-cell recording of $I_{Ca,L}$

The L-type calcium current ($I_{Ca,L}$) was measured in the whole-cell configuration of the patch clamp technique by an Axopatch 200B amplifier (Axon Instruments, Foster City, CA, USA) at 37°C. The experimental procedure followed a previous publication (Uzun *et al*, 2016). The ISO2 software was used for data acquisition and analysis (MFK, Niedernhausen, Germany). Heat-polished pipettes were used. Cell capacitance ($C_m$) was calculated from steady-state current during depolarizing ramp pulses (1 V/1 s) from −40 to −35 mV. $Ca^{2+}$ currents were elicited by applying test pulses from −80 to +10 mV (0.5 Hz). The cells were investigated in a small perfusion chamber placed on the stage of an inverse microscope. In order to avoid contaminating currents, $K^+$ currents were blocked by replacing $K^+$ with $Cs^+$ and tetraethylammonium chloride in the bath solution. The experiments were performed with the following $Na^+$-free bath solution (in mM): tetraethylammonium chloride 120, CsCl 10, HEPES 10, $CaCl_2$ 2, $MgCl_2$ 1, and glucose 20 (pH 7.4, adjusted with

CsOH) at 37°C. The pipette solution (pH 7.2, adjusted with CsOH) included (in mM): cesium methanesulfonate 90, CsCl 20, HEPES 10, Mg-ATP 4, Tris-GTP 0.4, EGTA 10, and $CaCl_2$ 3 (Christ *et al*, 2001). Current amplitude was determined as the difference between peak inward current and current at the end of the depolarizing step. The selective $I_{CaL}$-blocker nifedipine (10 μM) was used to identify $I_{Ca,L}$.

### Bioluminescence resonance energy transfer (BRET)

BRET assays were performed in HL-1 cardiac cells as previously described (Rusconi *et al*, 2016). Briefly, $Ca_v\alpha1.2$ and *ACTN2* (WT and mutant) cDNAs were cloned into the pNLF1-N and the HaloTag-pFN21A vector (Promega), respectively, and then transfected in HL-1 cells in a $Ca_v\alpha1.2$-NanoLuc:*ACTN2*-Halo (WT or mutant) 1:100 ratio. $Ca_v\beta2$ was added as indicated. Forty-eight hours after transfections, cells were treated with 100 nmol/l Nano-BRET 618 ligand, and signals were detected 5 h after treatment, with a Synergy 4 instrument (BioTek).

### Mathematical modeling and computer simulations of cardiomyocyte function

To simulate the effect of altered $I_{Ca,L}$ density on the AP and force development, we updated our previously used hiPSC-like computational CM model (Lemoine *et al*, 2018) to include the description of the contractile element, as implemented by Ji *et al* (2015) based on the original formulation from Negroni and Lascano (1996). The HCM-related change in the $I_{Ca,L}$ amplitude, which was observed in the patch clamp experiments, was accounted for by increasing the maximum conductance by 50% in the computational CM model. Simulation of electrophysiological and contractile function of the CM and post-processing was done with the commercial MATLAB software (release 2017b, https://www.mathworks.com/products/matlab.html), using the built-in function (ode15s) to solve the differential equations.

### Antibodies

All antibodies were commercially available, validated, and suitable for human species as specified by the manufacturer: Monoclonal anti-α-actinin used in a 1:800 dilution for immunofluorescence staining (Sigma, produced in mouse, Cat.: A7811-.2ML, Lot:027M4813V); goat anti-mouse IgG (H+L) highly cross-adsorbed secondary antibody; Alexa Fluor 488 used in a 1:800 dilution for immunofluorescence staining (Invitrogen, produced in goat, Cat.: A-11029, Lot:1829920); anti-ACTN2 used in a 1:1,000 dilution for Western blot analysis (Sigma, produced in rabbit, Cat.: SAB2108642-100UL, Lot: QC12269); anti-cardiac troponin T antibody used in a 1:5,000 dilution for Western blot analysis (Abcam, produced in mouse, Cat.: ab8295, Lot: GR3232545-1); anti-mouse IgG (whole molecule)–peroxidase antibody used in a 1:5,000 dilution for Western blot analysis (Sigma, produced in rabbit, Cat.: A9044-2ML, Lot: 055M4818V); anti-rabbit IgG (whole molecule)–peroxidase antibody used in a 1:6,000 dilution for Western blot analysis (Sigma, produced in goat, Cat.: A0545-1ML, Lot: 065M4769V); anti-cardiac troponin T-FITC used in a 1:10 dilution for FACS analysis (Miltenyi Biotec, Cat.: 130-119-575, Lot: 5181219151); and REA-control (I)-FITC used in a 1:10 dilution for

FACS analysis as isotype control (Miltenyi Biotec, Cat.: 130-104-611, Lot: 5180809130).

## Statistics

GraphPad Prism 5 and 8 (GraphPad Software, San Diego, CA, USA) was used for data analyses. Curves were fitted to data points from individual experiments, and exact number of replicates is indicated in figure legends. Data were compared using unpaired/paired Student's $t$-test, or using two-way ANOVA followed by Sidak's multiple comparisons test when appropriate, and for more than two groups one-way ANOVA followed by Bonferroni's post-test, if not indicated differently. All analyses were two-tailed, and a $P < 0.05$ was considered as statistically significant, whereby exact $P$-values are indicated in graphs. Values are expressed as mean ± SEM. For myofilament $Ca^{2+}$ sensitivity measurements, concentration–response curves were fitted to the data points and force–pCa relationship comparison was done by using extra sum-of-squares $F$-test.

# Data and materials availability

Data, analytic methods, and study materials will be made available on request to other researchers for purposes of reproducing the results or replicating the procedures within the limits required for patient confidentiality.

**Expanded View** for this article is available online.

## Acknowledgements

We are thankful to Alessandra Moretti (Technische Universität München, Munich, Germany) for providing the control hiPSC cell line. The authors gratefully acknowledge Umber Saleem, Ingra Mannhardt, Anika E. Knaust, Tessa Werner, Marta Lemme, Thomas Schulze, Birgit Klampe, Aya Domke-Shibamiya, Mirja Schulze, and Bärbel Ulmer (Pharmacology, Hamburg) for participating in the production of hiPSC-CM, Lukas Radziwolek (EACVI-certified) for blinded echocardiography analysis, Jörg Reinhardt (MedFact Engineering GmbH) for providing the MAP catheter and the FACS Core unit (Hamburg), Dr. Daniela Indenbirken (Technology platform next-generation sequencing, Heinrich Pette Institute, Hamburg) for performing RNA sequencing, Kristin Dawson for language editing and proofreading, and Suellen Lopes Oliveira for assistance in graphic design. This work was supported by the German Centre for Cardiovascular Research (DZHK) to M.D.L., A.Ha., and L.C.; the German Ministry of Research Education (BMBF) to M.D.L., A.Ha., and L.C.; the Deutsche Herzstiftung (F/15/17) to F.W.F; the Helmut und Charlotte Kassau Stiftung to L.C.; the German Research Foundation (DFG, 3423/5-1) to A.Ha.; the European Research Council Advanced Grant (IndivuHeart, 340248) to T.E.; the European Union's Horizon 2020 research and innovation program under the Marie Sklodowska-Curie grant (AFib-TrainNet, 675351) to T.E. and T.C.; the Research Promotion Fund of the Faculty of Medicine (Hamburg) to M.P. and M.D.L. ("Clinician Scientist Program" and "Project funding for young scientists"); and the Pirkanmaa Regional Fund of the Finnish Cultural Foundation and the Academy of Finland Centre of Excellence in Body-on-Chip Research to J.T.K.

## Author contributions

Conception and design of the work: MPr, MDL, TE, and LC; Methodology: MPr and MDL; Acquisition, analysis, and/or interpretation of the data: MPr, MDL, ATLZ, AH, VdM, JTK, NK, JB, TK, EK, SS, MS, FWF, JM, SDL, CR, AEV, AHa, GM, DC, CM, TC, MPa, TE, and LC; Patient recruitment: JM, MDL, and MPa; Original draft: MPr and MDL; Review and editing: MPr, MDL, DC, TC, TE, and LC; Funding acquisition, MPr, MDL, AHa, TC, TE, and LC. All authors participated in the critical review of the manuscript and approved the submitted version.

## Conflict of interest

A.Ha. and T.E. are cofounders of EHT Technologies GmbH, Hamburg. All the other authors declare no conflict of interest.

## For more information

(i) https://omim.org/entry/102573
(ii) https://www.mayoclinic.org/diseases-conditions/hypertrophic-cardiomyopathy/symptoms-causes/syc-20350198
(iii) https://www.heart.org/en/health-topics/cardiomyopathy/what-is-cardiomyopathy-in-adults/hypertrophic-cardiomyopathy
(iv) http://www.eht-technologies.com/
(v) https://dzhk.de/en/
(vi) https://www.ndr.de/ratgeber/gesundheit/UKE-Hamburg-Fortschritt-in-der-Herzforschung,herz588.html

## The paper explained

**Problem**

Hypertrophic cardiomyopathy (HCM) is the most prevalent inherited heart disease accompanied by structural and contractile alterations. The genetic diversity is paralleled by clinical heterogeneity, highlighting the need for individualized treatment approaches in HCM. The goal of this study was to elucidate the functional role of a novel mutation in *ACTN2*, encoding α-actinin 2, in human-induced pluripotent stem cell (hiPSC)-derived cardiomyocytes and translate *in vitro* findings to the affected family.

**Results**

We identified a rare *ACTN2* c.740C>T transition (p.T247M) in an HCM patient, her son, and sister. To identify underlying cellular pathologies, patient-derived hiPSC and CRISPR/Cas9-derived isogenic control were generated. Compared to control, HCM hiPSC-derived cardiomyocytes/engineered heart tissues recapitulated several hallmarks of HCM, such as myofibrillar disarray, hypercontractility, prolonged relaxation, and higher myofilament $Ca^{2+}$ sensitivity. Additionally, L-type calcium channel (LTCC) current density was higher and action potential duration (APD) was longer in HCM than in control. Mutant α-actinin 2 exhibited reduced interaction with the LTCC complex. The LTCC blocker diltiazem reduced force amplitude, relaxation, and APD to a greater extent in HCM than in control hiPSC-engineered heart tissue. We translated our findings to patient care and showed that diltiazem ameliorated the prolonged QTc interval in the HCM-affected son and sister of the index patient.

**Impact**

This study revealed a novel HCM mutation associated with a contractile and electrophysiological phenotype in hiPSC-derived cardiomyocytes and engineered heart tissues and a genotype-specific pathomechanism that guided clinical therapy in a family with HCM. Long-term follow-up is necessary to see whether diltiazem also prevents development of arrhythmia and progression of structural aspects of the disease. This study may serve as proof-of-principle for the use of hiPSC-cardiomyocytes and CRISPR/Cas9 for personalized treatment of cardiomyopathies.

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
