## [Review Process File · EMBO Molecular Medicine]

Disease modeling of an α -actinin 2 mutation guides clinical therapy in hypertrophic cardiomyopathy

Maksymilian Prondzynski, Marc D. Lemoine, Antonia T. L. Zech, András Horváth, Vittoria Di Mauro, Jussi T. Koivumäki, Nico Kresin, Josefine Busch, Tobias Krause, Elisabeth Krämer, Saskia Schlossarek, Michael Spohn, Felix W. Friedrich, Julia Münch, Sandra D. Laufer, Charles Redwood, Alexander E. Volk, Arne Hansen, Giulia Mearini, Daniele Catalucci, Christian Meyer, Torsten Christ, Monica Patten, Thomas Eschenhagen, Lucie Carrier

Review timeline:

Submission date:	3 July 2019
Editorial Decision:	30 July 2019
Revision received:	22 August 2019
Editorial Decision:	24 September 2019
Revision received:	8 October 2019
Accepted:	9 October 2019

Editor: Céline Carret

Transaction Report:

1st Editorial Decision

30 July 2019

Thank you for the submission of your manuscript to EMBO Molecular Medicine. We have now heard back from the three referees whom we asked to evaluate your manuscript.

As you will see from the set of comments pasted below, overall there is an agreement that the study is well executed and of interest. There are however some shortcomings as outlined by each of the reviewers. Therefore, we would like to encourage you to address the following: improve the discussion (ref. 1), check for the generality of the approach (ref. 1), tone down the diltiazem treatment conclusions (ref. 2 and 3), and add some RNAseq/transcriptomic analysis (ref. 2 and 3).

We would therefore welcome the submission of a revised version within three months for further consideration and would like to encourage you to address all the criticisms raised as suggested to improve conclusiveness and clarity. Please note that EMBO Molecular Medicine strongly supports a single round of revision and that, as acceptance or rejection of the manuscript will depend on another round of review, your responses should be as complete as possible.

Please also contact us as soon as possible if similar work is published elsewhere. If other work is published, we may not be able to extend the revision period beyond three months.

I look forward to receiving your revised manuscript.

***** Reviewer's comments *****

Referee #1 (Remarks for Author):

This is a very solid, right, and convincing study that shows clearly that information gleaned from hiPsc model system can lead to uncovering new therapeutic approaches for patients with a defined monogenic cardiac disease, namely HCM. Other studies have supported the utility of hiPsc models to understand disease, and to model key features of the phenotype, but, to my knowledge, none have taken this in a strictly personalized manner to show that there could be potential therapeutic benefit using surrogate indicators in the same patient. In many ways, this is an elegant "case study" at a molecular level, which is its strength and also its weakness. On the one hand, it shows the way forward for more personalized therapy using hiPsc model systems, and at the same time it is only indicative of a single mutation in a single family when it is clear the disease represents many personal mutations scattered across many sarcomeric genes. The question arises as to the generalizability of the finding. This point should be addressed in the discussion more completely. Also, there are personalized therapies for a distinct class of HCM due to mutations in myosin itself that are being tested by Myokardia. would these be expected to work in this patient or on the hiPsc model system? If not would lack of effect be also predictive? This would speak to the specificity of the response. If the authors believe that the diltiazem effect would apply to other forms of HCM, they could support this by examining one of the many other hiPsc lines that have been generated. Finally, the use of the novel 3-D muscle tissue system of Eschenhagen is a strength of this paper. It would be interest to highlight how the findings in this study may be different from other previous HCM iPsc models reported by Joe Wu and others.

Referee #2 (Comments on Novelty/Model System for Author):

The use of engineered heart tissues is outstanding and demonstrates than an ACTN2 mutation really does cause HCM. The use of the platform for drug screening is less solid.

Referee #2 (Remarks for Author):

The paper by Prondzynski and colleagues describes a small family with novel ACTN2 variant p.T247M, which is not found in large population databases. It has been suggested in the literature that the level of evidence for ACTN2 being a causative gene for hypertrophic cardiomyopathy is not strong. This work advances the evidence for ACTN2 being, at least for some variants, causing HCM. The authors generate iPSC derived cardiomyocytes from the proband with the ACTN2 p.T247M variant and find the cells are larger and have abnormal force and abnormal action potentials. The authors describe abnormal force/calcium relationships. They suggest that the Ltype calcium channel blocker diltiazem may be effective and treat less affected family members (n=2) with this agent and show reduction of the QT interval. The work around the ACTN2 variant and its role in HCM is quite good. This alone adds to the literature.

Major comment

1. The weakness of the work is around the suggestion that diltiazem and reduced QT interval is reliable indicator of outcomes. The presence of cardiomyopathy alone is well known to prolong QT intervals and is not felt to be a primary finding. Furthermore, reducing the QT interval in this setting is not of clear clinical value. All the wording and conclusions around using diltiazem to treat arrhythmias and long QT should be softened since this is not a clinical trial and the number of treated subjects is simply too small and without adequate outcomes to make this assertion. Furthermore, the data linking the L type channel to α -actinin is quite limited. If there is an effect on calcium handling and action potentials, it is likely quite downstream and much more would need to be shown to assert a more proximal relationship.

2. Allelic balance is better assessed using RNAseq rather than a subcloning method that evaluates very few clones.

Minor comments

3. It is worthwhile to emphasize that this is a later onset HCM, compared to some genetic mutations which can present quite early in life in young adults or even children. Is there any clinical history on the parents of 1-2 to know which side of the family it derived? Any early death in either paternal or maternal side?
4. Was the 43 yo affected individual screened for abnormal rhythms, if so how? Is there more data on this patient? MRI findings- details?
5. The clinical descriptions need to be rewritten since there is some unusual and non-native English here.

Referee #3 (Comments on Novelty/Model System for Author):

The manuscript by Prondzynski et al. reports on a novel HCM mutation in alpha-Actinin-2, which is associated with hypertrophic cardiomyopathy (HCM). The authors are using an impressive amount of different techniques and experimental approaches to characterize the pathological phenotype present in the index patient using iPS cell derived cardiac myocytes, which are either cultured in 2 dimensional cultures or as EHT tissue model. The authors also control for genetic background effects by rescuing the mutation by CRISPR Cas9-mediated homology repair demonstrating a partial rescue of the phenotype. Given the importance of heart failure as a prevalent disease and the importance to develop novel patient-specific therapeutic approaches, this manuscript details how one can come up with novel insight, which guided the researchers to propose some novel treatment regime, which had some impact on the pathological phenotype. iPS cells, rescuing the genotype homology repair, and careful phenotype analysis are executed to a very high standard in this manuscript.

Referee #3 (Remarks for Author):

The length of the manuscript is appropriate and I think that overall the paper is extremely well-written, and the discussion is focused on the most important aspects.

1. What I am struggling with is the intermediate phenotype of the rescued iPS cell line. If the ACTN2 mutant would be the only disease-causing mutation present in this patient would you not expect a more complete rescue. Several measured parameters as for example cell area, which reflects something like a prohypertrophic state is clearly intermediate to the mutant but also clearly higher than in WT. The same holds true for force, relaxation etc.
2. I am not a geneticist, so it is hard for me to decide whether maybe another coupled pathogenic allele could be present, which acts in combination with ACTN2 to cause the phenotype. You have tested a couple of candidates but clearly a transcriptome analysis and a comparison of affected and normal family members would maybe give some information what else is causing this intermediate pathology after ACTN2 rescue.
3. How is the mutation in ACTN2 mechanistically linked to a gain of function of the L-type calcium channel. Can you enlighten the reader how this is mechanistically linked? What is known to cause a similar gain of function?
4. Although diltiazem was able to lower QT time, it is still far away from normalizing the QT time. This suggest that maybe other elements controlling calcium transients are probably also aberrantly modulated in cells transcribing the mutant allele.

1st Revision - authors' response

22 August 2019

Response to Referee #1

We would like to thank the reviewer for his/her careful work on our manuscript and his/her nice, positive comments. We have carefully considered your comments and modified the discussion accordingly.

Please note that all changes in the manuscript are marked with yellow.

This is a very solid, right, and convincing study that shows clearly that information gleaned from hiPSc model system can lead to uncovering new therapeutic approaches for patients with a defined

monogenic cardiac disease, namely HCM. Other studies have supported the utility of hiPS models to understand disease, and to model key features of the phenotype, but, to my knowledge, none have taken this in a strictly personalized manner to show that there could be potential therapeutic benefit using surrogate indicators in the same patient. In many ways, this is an elegant "case study" at a molecular level, which is its strength and also its weakness. On the one hand, it shows the way forward for more personalized therapy using hiPSc model systems, and at the same time it is only indicative of a single mutation in a single family when it is clear the disease represents many personal mutations scattered across many sarcomeric genes. The question arises as to the generalizability of the finding. This point should be addressed in the discussion more completely. Also, there are personalized therapies for a distinct class of HCM due to mutations in myosin itself that are being tested by Myokardia. Would these be expected to work in this patient or on the hiPSc model system? If not would lack of effect be also predictive? This would speak to the specificity of the response. If the authors believe that the diltiazem effect would apply to other forms of HCM, they could support this by examining one of the many other hiPSc lines that have been generated. Finally, the use of the novel 3-D muscle tissue system of Eschenhagen is a strength of this paper. It would be interest to highlight how the findings in this study may be different from other previous HCM iPSc models reported by Joe Wu and others.

Response: We thank the reviewer for his/her positive comments. We agree that this is a "case study" evaluating a single HCM mutation out of more than 1000 identified in HCM. However, and specifically, the level of evidence for ACTN2 being a causative gene in HCM is not strong as compared to MYBPC3 and MYH7 genes (Walsh R et al., Genet Med 2017). Furthermore, the interesting fact that the prolonged QTc did not find attention during clinical routine but was revealed after the work in vitro underlines the necessity for more personalized medicine. Therefore, using isogenic control and different assays in vitro we validated its causal role in HCM and could change the treatment of the patient. We modified the discussion to better stress the "case study" and the fact that other less evident genes could be validated by this approach (page 9, lines 264-265; page 10, lines 334 and 336; page 11, line 337 and lines 340-342).

The Myocardia compound MYK-461 (mavacamten) is a myosin inhibitor that stabilizes the super-relaxed state (SRX) of myosin and normalized hypercontractility associated with MYH7 and MYBPC3 mutations (Anderson RL et al., PNAS 2018; Toepfer CN et al., Sci Transl Med 2019). Therefore, it could also work for the ACTN2 mutation, which was associated with higher force in EHTs. However, this compound would not be expected to ameliorate the larger LTCC current detected in HCM-cardiomyocytes, which we believe is the primary cause for the prolonged QTc interval observed in the family members carrying the ACTN2 mutation. Thus, the iPSC-approach revealed a mechanism offering a more specific therapeutic intervention (with diltiazem).

Diltiazem has been very recently used in healthy or HCM iPSC-cardiomyocytes by the group of Joe Wu (Lam CK et al., Circ Res 2019; Wu H et al., Eur Heart J 2019). They showed that diltiazem ameliorates HCM-phenotypes and particularly abnormal relaxation in 2Dcultured hiPSC-cardiomyocytes carrying MYH7, MYBPC3 or TNNT2 missense mutations (Wu H et al., 2019 Eur Heart J). These data are in line with our findings (relaxation), but did not reveal the underlying mechanism. The data are now discussed (page 10, lines 306-308).

Finally, we also believe that the EHT model was crucial for modeling contractile HCMphenotypes and for evaluating diltiazem as a treatment option in this study. We are currently not aware of any hiPSC 3D model that has been used to investigate disease-specific phenotypes for HCM except the EHT model (Smith JGW et al., Stem Cell Rep 2018; Mosqueira D et al., Eur Heart J 2018). Nevertheless, besides the EHT model we believe that the next most commonly used 3D model is the engineered human myocardium (EHM) used by the Zimmermann group in Göttingen (Tiburcy M et al., Circulation 2017) and adapted by Joe Wu's group (Lam CK et al., Circ Res 2019). The recent publication of this group tested diltiazem and other LTCC blockers in EHTs from healthy control cardiomyocytes with similar results as observed in our study, namely reduction in force and relaxation. This is what one would expect from a LTCC blocker. The point we make in our study is the larger (and as we believe specific) effect of diltiazem in ACTN2-mutated than control EHTs. Extensive electrophysiological studies have not been undertaken for the EHM as they were done for EHTs (Uzun AU et al., Front Physiol 2016; Lemoine MD et al., Sci Rep 2017; Horvath A et al. 2018 Stem Cell Reports; Lemoine MD et al., Circ Arrhythm Electrophysiol 2018). Therefore, EHTs are an important model for investigating electrophysiological components in 3D, which is not

commonly done in the tissue engineering community.

Response to Referee #2

We would like to thank the reviewer for his/her comments and critiques on our manuscript. We have softened some of our wordings and performed an additional experiment to address your point. Please note that all changes in the manuscript are marked with yellow.

The paper by Prondzynski and colleagues describes a small family with novel ACTN2 variant p.T247M, which is not found in large population databases. It has been suggested in the literature that the level of evidence for ACTN2 being a causative gene for hypertrophic cardiomyopathy is not strong. This work advances the evidence for ACTN2 being, at least for some variants, causing HCM. The authors generate iPSC derived cardiomyocytes from the proband with the ACTN2 p.T247M variant and find the cells are larger and have abnormal force and abnormal action potentials. The authors describe abnormal force/calcium relationships. They suggest that the L-type calcium channel blocker diltiazem may be effective and treat less affected family members (n=2) with this agent and show reduction of the QT interval. The work around the ACTN2 variant and its role in HCM is quite good. This alone adds to the literature.

Major comment

1. The weakness of the work is around the suggestion that diltiazem and reduced QT interval is reliable indicator of outcomes. The presence of cardiomyopathy alone is well known to prolong QT intervals and is not felt to be a primary finding. Furthermore, reducing the QT interval in this setting is not of clear clinical value. All the wording and conclusions around using diltiazem to treat arrhythmias and long QT should be softened since this is not a clinical trial and the number of treated subjects is simply too small and without adequate outcomes to make this assertion. Furthermore, the data linking the L type channel to α -actinin is quite limited. If there is an effect on calcium handling and action potentials, it is likely quite downstream and much more would need to be shown to assert a more proximal relationship.

Response: We thank the reviewer for this comment. We agree that cardiomyopathy as such can be associated with prolonged QT intervals and this cannot be excluded as a contributor in the case of this mutation. However, we provide evidence for a more specific effect of the actinin mutation on the LTCC current and believe that the larger diltiazem effect in mutation-carrier EHTs and the new data on the interaction of wild-type and mutated actinin with Cav α 1.2 (see below) support this view.

We are fully aware of the fact that shortening of the QT interval in the HCM patients as such is of unknown therapeutic significance. We apologize if we used too strong wording and softened the wording in the abstract, results and discussion, accordingly (page 2, lines 47 and 48; page 7, line 212; page 8, line 236; page 10 lines 330-333).

We also agree that the link between ACTN2 and L-type calcium channel was not fully elucidated. And since reviewer #3 also asked for mechanistic insights, we performed an additional experiment in collaboration with the group of Daniele Catalucci (Milan, Italy). Indeed, it has been shown that α -actinin 2 interacts with ion channels and contributes to their modulation, such as the L-type calcium channel (LTCC) complex. In particular, by binding to the IQ segment of the Cav α 1.2, pore unit of the LTCC, α -actinin 2 modulates LTCC density and function at the plasma membrane. Therefore, we tested whether the α -actinin 2 mutation affects the binding of α -actinin 2 to Cav α 1.2. By a bioluminescence resonance energy transfer (BRET) assays performed in live cardiac myocyte-like HL-1 cell, we observed a similar binding affinity of Cav α 1.2 to WT and mutant α -actinin 2 (new Figure S4). However, while this binding affinity of Cav α 1.2 to WT α -actinin 2 strongly increased with the co-transfection of Cav α 2, the LTCC accessory subunit and chaperone of the LTCC pore unit, it did not in case of mutant α -actinin 2. Thus, the HCM mutation decreased the interaction of α -actinin 2 with the LTCC complex, possibly affecting the activity of the channel. This may explain the electromechanical phenotype of the investigated HCM-affected family, characterized by higher I_{Ca,L} density. This experiment has been added in the revised version of the manuscript and discussed (Results, page 6, line 175; page 7, lines 196-210; Discussion, pages 10-11, lines 291-301 and line 347; Methods, page 19, lines 621-628).

2. Allelic balance is better assessed using RNAseq rather than a subcloning method that evaluates

very few clones.

Response: Thank you for the suggestion. We performed RNAseq in different samples and confirmed the presence of both wild-type and mutant mRNA in a 50:50 distribution in both HCM-EHTs and in septal myectomy of patient II.4, confirming the allelic balance (see new Figure S2C; Results page 5, line 143; Methods, page 16, lines 509-526).

Minor comments

3. It is worthwhile to emphasize that this is a later onset HCM, compared to some genetic mutations which can present quite early in life in young adults or even children. Is there any clinical history on the parents of I-2 to know which side of the family it derived? Any early death in either paternal or maternal side?

Response: We agree with the reviewer and modified the clinical characterization accordingly (page 4, lines 121-123). Furthermore, we do not have any information from the parents of individual I.2, which could indicate transmission from either the father or the mother.

4. Was the 43 yo affected individual screened for abnormal rhythms, if so how? Is there more data on this patient? MRI findings- details

Response: We added long ECG and MRI examinations of the patient III.4 (43 yo) and added this information in the revised manuscript (Results, page 4, lines 110-114).

5. The clinical descriptions need to be rewritten since there is some unusual and non-native English here.

Response: Thank you for notifying it. We asked an English native speaker to correct the clinical description of the family (page 4, lines 102-123).

Response to Referee #3

We would like to thank the reviewer for his/her careful work on our manuscript, the positive feedback and constructive comments and suggestion to improve the manuscript. We have performed new experiments to address your points. Please note that all changes in the manuscript are marked with yellow.

The manuscript by Prondzynski et al. reports on a novel HCM mutation in alpha-Actinin-2, which is associated with hypertrophic cardiomyopathy (HCM). The authors are using an impressive amount of different techniques and experimental approaches to characterize the pathological phenotype present in the index patient using iPS cell derived cardiac myocytes, which are either cultured in 2 dimensional cultures or as EHT tissue model. The authors also control for genetic background effects by rescuing the mutation by CRISPR Cas9-mediated homology repair demonstrating a partial rescue of the phenotype. Given the importance of heart failure as a prevalent disease and the importance to develop novel patient-specific therapeutic approaches, this manuscript details how one can come up with novel insight, which guided the researchers to propose some novel treatment regime, which had some impact on the pathological phenotype. iPS cells, rescuing the genotype homology repair, and careful phenotype analysis are executed to a very high standard in this manuscript. The length of the manuscript is appropriate and I think that overall the paper is extremely well-written, and the discussion is focused on the most important aspects.

1. What I am struggling with is the intermediate phenotype of the rescued iPS cell line. If the ACTN2 mutant would be the only disease-causing mutation present in this patient would you not expect a more complete rescue. Several measured parameters as for example cell area, which reflects something like a prohypertrophic state is clearly intermediate to the mutant but also clearly higher than in WT. The same holds true for force, relaxation etc.

Response: We thank the reviewer for the attentive hint. Indeed, several parameters showed intermediate values for the isogenic control cell line in between the HCM and the control cell lines. It could indeed indicate an incomplete rescue with CRISPR/Cas9 or genetic variability between cell lines. However, all parameters were significantly different between the HCM and the isogenic control line carrying the same genetic background, providing evidence that the mutation is

responsible of the disease phenotype. On the other hand, our control cell line is not the universal reference and variability between control cell lines has been reported in several studies. For example, it has been recently shown that response to drugs varied between different control cell lines (for example to diltiazem, Lam CK et al., *Circ Res* 2019). We also previously found significantly different APD90 in 3 different healthy hiPSCcardiomyocyte cell lines (180 ms vs. 240 ms vs 290 ms; Lemoine et al., *Circ Arrhythm Electrophysiol* 2018, Supplemental Figure III b). Finally, we have systematically analysed in a blinded manner 36 control cell lines, which showed that the the SD of EHT parameters are 20-35% of the mean (Eschenhagen, personal communication; see results below). All this underlines the need of isogenic control for personalized medicine. We therefore discussed further and added the respective references (page 9, lines 275-286).

Parameters obtained under 1 Hz-pacing in 36 control iPSC-derived engineered heart tissues (Mean \pm SD):

Force = 0.151 ± 0.053 mN (\pm 35% of the mean)

T180% = 0.143 ± 0.029 sec (\pm 20% of the mean)

T280% = 0.205 ± 0.069 sec (\pm 34% of the mean)

2. I am not a geneticist, so it is hard for me to decide whether maybe another coupled pathogenic allele could be present, which acts in combination with ACTN2 to cause the phenotype. You have tested a couple of candidates but clearly a transcriptome analysis and a comparison of affected and normal family members would maybe give some information what else is causing this intermediate pathology after ACTN2 rescue.

Response: In the line of the response to your comment 1, we think that the main genetic difference is between the control and isogenic control, but we do not have DNA samples of the control individual to confirm that. You suggest performing a transcriptome analysis of affected and normal family members, but we cannot get cardiac samples from unaffected individuals. Thus, we think it will be difficult to solve this issue with RNAseq with patient's tissue. Further OMIC analyses are planned on several HCM and isogenic control cell lines but will be part of another study. On the other hand, and to respond in part to your question, preliminary RNAseq data obtained on HCM, HCMrep and Ctrl EHTs showed the following correlation of gene expression (counts). This shows a higher variability between HCMrep and Ctrl EHTs than HCMrep and HCM EHTs, underlying more genetic variability between the two control cell lines than between the HCM and isogenic control lines (n=7 EHTs out of 3-6 batches of cardiomyocyte differentiation).

3. How is the mutation in ACTN2 mechanistically linked to a gain of function of the L-type calcium channel. Can you enlighten the reader how this is mechanistically linked? What is known to cause a similar gain of function?

Response: thank you for your very interesting point, which we also tried to understand. It has been shown that α -actinin 2 interacts with ion channels and contributes to their modulation, such as the L-type calcium channel (LTCC) complex. In particular, by binding to the IQ segment of the Cav α 1.2, pore unit of the LTCC, α -actinin 2 modulates LTCC density and function at the plasma membrane. Therefore, we tested whether the α -actinin 2 mutation affects the binding of α -actinin 2 to Cav α 1.2. By a bioluminescence resonance energy transfer (BRET) assays performed in live cardiac myocyte-like HL-1 cells we observed a similar binding affinity of Cav α 1.2 to WT and mutant α -actinin 2 (new Figure S4). However, while this binding affinity of Cav α 1.2 to WT α -actinin 2 strongly increased with the co-transfection of Cav α 2, the LTCC accessory subunit and chaperone of the LTCC pore unit, it did not in case of mutant α -actinin 2. Thus, the HCM mutation decreased the interaction of α -actinin 2 with the LTCC complex, possibly affecting the activity of the channel. This may explain the electromechanical phenotype of the investigated HCM-affected family, characterized by higher I_{Ca,L} density. This experiment has been added in the revised version of the manuscript and discussed (Results, page 6, line 175; page 7, lines 196-210; Discussion, pages 9-10, lines 291-301 and line 348; Methods, page 19, lines 621-628).

2nd Editorial Decision

24 September 2019

Thank you for the submission of your revised manuscript to EMBO Molecular Medicine. We have now received the enclosed reports from the referees that were asked to re-assess it. As you will see the reviewers are now globally supportive and I am pleased to inform you that we will be able to accept your manuscript pending minor editorial amendments.

***** Reviewer's comments *****

Referee #2 (Remarks for Author):

None

Referee #3 (Comments on Novelty/Model System for Author):

The use of iPS cell model and to further differentiate them in 3D culture is an advanced model of analysing the pathomechanisms that cause HCM related phenotypes. This is highly innovative and exemplifies state of the art of modelling heart disease with iPS cells.

Referee #3 (Remarks for Author):

The authors have addressed my points of concerns and even performed some additional experiments, which demonstrate a biochemical difference between the mutant and wildtype α -actinin and provides an functional explanation for the effect of the mutation on LTCC.

2nd Revision - authors' response

8 October 2019

Authors made the requested editorial changes.

Corresponding Author Name: Lucie Carrier

Journal Submitted to: EMBO Mol Med

Manuscript Number: EMM-2019-11115